# Hyper-Modality Enhancement for Multimodal Sentiment Analysis with Missing Modalities

**Yan Zhuang**[1,†], **Minhao Liu**[1,2,†], **Wei Bai**[3], **Yanru Zhang**[1,2], **Wei Li**[1]
**Jiawen Deng**[1,*], **Fuji Ren**[1,2,*]
[1]University of Electronic Science and Technology of China
[2]Shenzhen Institute for Advanced Study, UESTC, [3]Tsinghua University
{202211081370,202311081539}@std.uestc.edu.cn, bw23@mails.tsinghua.edu.cn
{minhaoliu,yanruzhang,dengjw,renfuji}@uestc.edu.cn
[†]Equal contribution [*]Corresponding authors

## Abstract

Multimodal Sentiment Analysis (MSA) aims to infer human emotions by integrating complementary signals from diverse modalities. However, in real-world scenarios, missing modalities are common due to data corruption, sensor failure, or privacy concerns, which can significantly degrade model performance. To tackle this challenge, we propose **Hyper-Modality Enhancement (HME)**, a novel framework that avoids explicit modality reconstruction by enriching each observed modality with semantically relevant cues retrieved from other samples. This cross-sample enhancement reduces reliance on fully observed data during training, making the method better suited to scenarios with inherently incomplete inputs. In addition, we introduce an **uncertainty-aware fusion mechanism** that adaptively balances original and enriched representations to improve robustness. Extensive experiments on three public benchmarks show that HME consistently outperforms state-of-the-art methods under various missing modality conditions, demonstrating its practicality in real-world MSA applications.

## 1 Introduction

Multimodal Sentiment Analysis (MSA) aims to identify an individual's emotional state toward a given topic, person, or entity by leveraging complementary signals from modalities such as language, audio, and video [1–3]. By integrating verbal, vocal, and visual cues, MSA offers a more nuanced understanding of human emotions and has been widely applied in emotion-aware systems like human-computer interaction and social media analysis. Considerable progress has been made in designing sophisticated fusion networks to integrate multimodal information[4–9]. However, in real-world scenarios, missing modalities frequently occur due to factors such as data corruption, sensor failures [10], or privacy concerns[11, 12], which pose significant challenges to the robustness of MSA systems.

Recent efforts have explored the challenge of MSA under missing modality conditions [13–18]. For example, IMDer [15] adopts a diffusion-based model to reconstruct missing modalities, while HRLF [18] employs a teacher-student framework, where the teacher model is trained with fully observed data and the student model learns to handle missing modalities by knowledge transfer. Although these methods achieve promising results, they share two key limitations. (i) They typically adopt a **pseudo-missing setting**, where missing modalities are artificially simulated by masking full-modality training data, while the ground-truth representations of the masked modalities are still used for supervision. However, such fully observed samples are often scarce in practice, limiting the scalability and real-world applicability of these methods. (ii) These approaches mainly rely on

the modalities available in the current sample, overlooking the potential of leveraging cross-sample information that could enrich the representations and enhance the robustness of sentiment predictions.

To address these challenges, we propose Hyper-Modality Enhancement (HME), a framework designed to address missing modalities without relying on full-modality reconstruction. Specifically, HME consists of two key components: (i) **Hyper-Modality Representation Generation**: This component learns a shared representation for the current sample using a small set of learnable prompts, conditioned on its available modalities. Simultaneously, it extracts modality-specific information from other samples to enrich this shared representation with sentiment-relevant signals. To ensure robustness, we apply a Variational Information Bottleneck (VIB) [19], which filters out irrelevant or noisy information. (ii) **Uncertainty-aware Fusion**: Given that representations from different modalities and samples may vary in reliability, HME incorporates an uncertainty-aware fusion mechanism that estimates the confidence of each input. This mechanism adaptively integrates these representations, mitigating the influence of noisy or unreliable signals and enhancing robustness under missing modality conditions. Together, these components mitigate the limitations of pseudo-missing training strategies and enhance the model's robustness in coping with realistic missing modality patterns. Our main contributions are as follows.

- A novel hyper-modality enhancement mechanism is introduced to leverage cross-sample semantic cues for strengthening observed modalities, without relying on explicit modality reconstruction.

- An uncertainty-aware fusion mechanism is developed to adaptively combine original and enriched features, enabling robust sentiment prediction under incomplete modality conditions.

- Our extensive experiments on three benchmark datasets demonstrate that HME achieves superior performance and greater robustness compared to existing methods.

## 2 Related Work

### 2.1 Multimodal Sentiment Analysis with Missing Modalities

Multimodal Sentiment Analysis (MSA) aims to predict sentiment by integrating information from language, video, and audio modalities. However, many established approaches [5, 7, 20–23] assume the consistent availability of all modalities across training, validation, and testing. Their performance often degrades significantly when any modality is missing [14, 15, 18], a scenario frequently encountered in real-world applications. To tackle this challenge, some approaches attempt to reconstruct missing modalities from the observed ones within the same sample like DiCMoR [24], IMDer [15], GCNet [13], MCTN [25], MMIN [26], DCCA [27], and DCCAE [28], often relying on ground-truth representations of the masked modalities for supervision. Others adopt a teacher-student framework, where a teacher model trained on full-modality data guides a student model operating on incomplete inputs [16–18, 11]. Although effective when trained on fully observed datasets, these methods commonly assume access to complete data during training to supervise either the reconstruction or the knowledge transfer process. This assumption limits their applicability in scenarios where full-modality samples are scarce or unavailable. Besides, existing methods often treat each sample independently, overlooking valuable contextual cues that could be leveraged from semantically related instances. In contrast, our proposed HME framework operates directly on incomplete data, without relying on modality reconstruction or teacher supervision. It enhances representations by leveraging contextual information from related instances, enabling more robust sentiment prediction.

### 2.2 Hyper-Modality Representation

Recent advancements in multimodal learning have explored "hyper-modality representations" [21, 14] to efficiently integrate information from diverse sources. These approaches often employ compact prompts—distilled representations of individual modalities—to capture task-relevant features. For instance, MPLMM [29] pre-trains on complete multimodal datasets, distilling this comprehensive knowledge into prompts that can later supplement or stand in for missing modalities. Similarly, methods like ALMT [21] and LNLN [14] construct hyper-modality representations by distilling information from other modalities into prompts, but crucially, they presume the constant availability of the language modality as a primary anchor. While these techniques demonstrate strong performance,

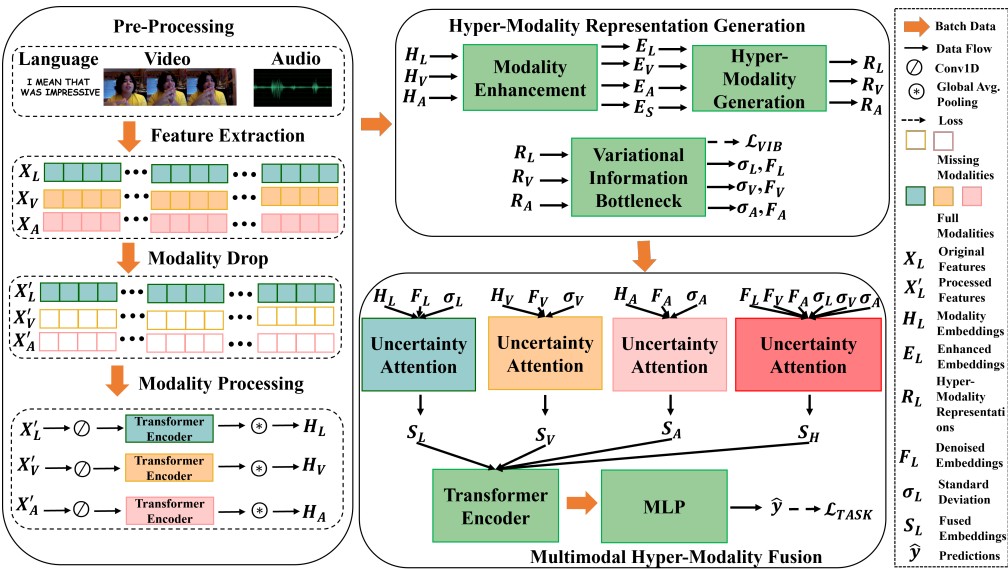

Figure 1: Visualization of HME framework. It consists of three modules: Pre-Processing Module, Hyper-Modality Representation Generation Module and Multimodal Hyper-Modality Fusion Module.

their practical applicability is often limited by their core assumptions: either the necessity of pre-training on complete datasets or the continuous presence of a specific modality. Such requirements are frequently unrealistic in real-world scenarios. In contrast, our proposed HME model is designed to robustly generate hyper-modality representations directly from any available modality, without these restrictive prerequisites.

## 3 Methodology

The overall structure of the proposed HME is shown in Figure 1. It consists of three modules, the Pre-Processing module processes the multimodal input to extract the modality-specific representations (introduced in Section 3.2), the Hyper-Modality Representation Generation Module provides enhanced hyper-modality information (introduced in Section 3.3), and Multimodal Hyper-Modality Fusion module fuses the enhanced representations with original representations based on the uncertainty of the enhanced representations (introduced in Section 3.4).

### 3.1 Problem Definition

Given the multimodal input $D = \{X_L^i, X_A^i, X_V^i\}_{i=1}^N$, each representation $X_m^i \in \mathbb{R}^{T_m \times d_m}$ is characterized by a sequence length $T_m$ and feature dimensionality $d_m$, where $m \in \{L, V, A\}$ denotes modality and $N$ is the length of the dataset. For simplicity, we omit the superscript $i$ and use $X_m$ to denote the representation of modality $m$. In real-world scenarios, some modalities may be missing for individual samples. To model this, missing modality representations are replaced with zero vectors [16, 13, 14]. For simplicity, we use $X_m'$ to uniformly represent the modality $m$ regardless of whether it is present or not. The goal of MSA with missing modalities is to predict the sentiment $\hat{y}$ using incomplete modalities $\{X_L', X_V', X_A'\}$.

### 3.2 Pre-Processing Module

Representations of each sample are firstly extracted to form the initial modality representation $\{X_L, X_V, X_A\}$. To simulate real-world scenarios with incomplete data, random modality dropout is applied to generate inputs with missing modalities, denoted as $\{X_L', X_V', X_A'\}$. For each input modality $X_m', m \in \{L, V, A\}$, following prior research [16, 22], we then use a 1D temporal convolutional layer with a kernel size of $3 \times 3$ to standardize all modalities to the same dimension $(d)$ and sequence length $(T)$. Position embeddings [30] are added to encode temporal information, resulting

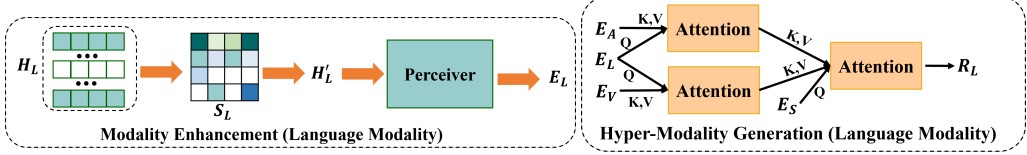

(a) Modality Enhancement module.    (b) Hyper-Modality Generation module.

Figure 2: The pipeline of Modality Enhancement (a) and Hyper-Modality Generation (b) module (e.g. using language modality). Here 'Q, K, V' denote the query, key and value in attention, respectively.

in: $\hat{X}_m = W_{3\times3}(X'_m) + PE(T, d)$, where $PE(T, d)$ denotes position embeddings. These representations are passed through a transformer encoder [30], $\mathcal{F}^p_{\phi_m}(\cdot)$, to capture dynamic modality-specific information. Finally, the Global Average Pooling (GAP) layer is applied to produce compact modality representations: $H_m = GAP(\mathcal{F}^p_{\phi_m}(\hat{X}_m))$, where $H_m \in \mathbb{R}^d$, $\phi_m$ is the network parameter.

### 3.3 Hyper-Modality Representation Generation Module

The Hyper-Modality Representation Generation Module generates the enriched modality representation by injecting sentiment-relevant information from other samples. Given modality-specific representations $H_m$, the module sequentially applies three components: Modality Enhancement, Hyper-Modality Generation and Variational Information Bottleneck, respectively. First, Modality Enhancement strengthens each modality's and each sample's representations via a few learnable prompts, enabling prompts to dynamically gather informative signals from samples in current batch. Then, Hyper-Modality Generation fuses the enhanced modality specific and shared information, selectively integrating cross-sample sentiment cues into a richer, hyper-modality representation. Finally, Variational Information Bottleneck filters noise and redundancy, distilling compact, high-quality representations. We next detail each sub-module.

**Modality Enhancement.** For each modality $m$, a similarity matrix is first computed using samples in the current batch, denoted as $S_m \in \mathbb{R}^{B\times B}$, where $B$ is the batch size. Taking modality $L$ as an example in Figure 2 (a), the similarity between representations is measured using cosine similarity [16, 31]: $sim(h_i, h_j) = \frac{h_i^\top h_j}{||h_i||\cdot||h_j||}$, where $h_i$ and $h_j$ denote different samples within the same modality $m$ in current batch. To better handle missing modalities and to reduce the reliance on individual representations—which can introduce noise—while ensuring that the selected representations remain meaningful, we simultaneously apply a similarity threshold and averaging strategy. Specifically, a similarity threshold $t_s$ is set: representations with similarity scores exceeding $t_s$ are selected and then averaged to form an enhanced representation. If no representation satisfies the threshold $t_s$, or if the modality is missing, we instead average all available representations of the current modality within the batch to capture the modality-specific information. This process is formulated as:

$$H'_{m,i} = \begin{cases} Avg(H_m(Idx)), & if\, Idx \neq \varnothing \\ Avg(H_{m,j} \neq 0), & otherwise \end{cases},\tag{1}$$

where $Idx = [S_m(i, j) > t_s]$ denotes the indexes of the representations, of which the similarities with $i$-th representation are higher that $t_s$. And $H_{m,j} \neq 0$ denotes the non-missing representations of modality $m$. $Avg(\cdot)$ denotes averaging the representations.

After obtaining the enhanced representations $H'_m$, Perceiver model [32, 21, 14] is employed to refine these representations. Specifically, the Perceiver model uses learnable prompts to learn from the representations by using cross-modal transformer blocks and transformer blocks. In the cross-modal transformer, the input representations $H'_m$ serve as keys and values, while learnable prompts $E'_m \in \mathbb{R}^{l_p \times d}$ act as queries. Here $l_p$ is the length of the prompt. And the attention [30] is defined as:

$$E_m = ATTN_{\phi_{m,m}}(E'_m, H'_m) = \text{Softmax}(\frac{E'_m W_{Q_e} W_h^\top (H'_m)^\top}{\sqrt{d}})H'_m W_{V_h},\tag{2}$$

where $W_{Q_e}, W_h^\top, W_{V_h} \in \mathbb{R}^{d\times d}$ are trainable weights. Subsequent layers apply a standard transformer structure with identical input for the query, key, and value. The final enhanced modality representation $E_m \in \mathbb{R}^d$ is obtained by averaging the output of the last transformer layer.

To capture common information across modalities and facilitate hyper-modality interaction, we introduce another set of learnable prompts $E'_S \in \mathbb{R}^{l_p \times d}$. These prompts are designed to aggregate features from all available modalities for each sample, even when some modalities are missing. The computation of $E_S$ follows a similar procedure to Eq. 2, with three key modifications: (i) replacing $E'_m$ with $E'_S$; (ii) replacing $H'_m$ with the concatenated features $[H_L, H_V, H_A]$; and (iii) adjusting the dimensions of the associated trainable weights accordingly. The final representation $E_S \in \mathbb{R}^d$ is also obtained by averaging the output of the last transformer layer.

**Hyper-Modality Generation.** This module generates the representation of each modality of the current sample by integrating sentiment-relevant information from other samples, thus obtaining the hyper-modality representation, which can be seen in Figure 2 (b). Taking modality $L$ as an example, we compute the enhanced representations of modality $L$ with modalities $V$ and $A$ using Eq. 2 by setting queries to $E_L$, keys and values to $E_A$ and $E_V$. These are defined as $E_{L,V} = ATTN_{\phi_{L,V}}(E_L, E_V)$ and $E_{L,A} = ATTN_{\phi_{L,A}}(E_L, E_A)$. The sentiment-relevant information are then distilled into the overall representation of each sample through setting $E_S$ as query: $R_L = ATTN_{\phi_{S,L}}(E_S, [E_{L,V}, E_{L,A}])$. Through performing on all three modalities, we obtain $R_m = ATTN_{\phi_{S,m}}(E_S, [E_{m,m_1}, E_{m,m_2}])$, where $m, m_1, m_2 \in \{L, V, A\}, m \neq m_1 \neq m_2$.

**Variational Information Bottleneck.** Selecting representations based solely on similarity can introduce errors, as samples with similar features may have different labels. Furthermore, the uncertain availability of modalities can also lead to redundancy when integrating information. To address these challenges, we employ the Variational Information Bottleneck (VIB) [19] to reduce noise and redundancy in the representations. VIB approximates the information bottleneck [33–35] by learning an encoding $Z$ that is maximally expressive about target $Y$ while being maximally compressive about input $X$ through optimizing the following function:

$$\mathcal{L}_{VIB} = I(Z, Y) - \beta I(Z, X). \tag{3}$$

Here $I(\cdot)$ is the mutual information, $\beta$ is the Lagrange multiplier. Specifically, we follow the previous ways that use the $\mathcal{L}_{TASK}$ (introduced in Eq. 9) and Kullback Leibler divergence [19, 36] to measure the $I(Z, Y)$ and $I(Z, X)$ through:

$$\mathcal{L}_{VIB} = \frac{1}{N} \sum_{i=1}^{N} \mathcal{L}_{TASK}(\hat{y}_i^{ib}, y_i) + \beta KL(p(z_i|x_i)||\mathcal{N}(0, \mathbf{I})), \tag{4}$$

where: $p(z_i|x_i) \sim \mathcal{N}(\mu_i, \sigma_i^2 \mathbf{I})$, and $\mu_i = f_{\theta_1}(x_i)$, $\sigma_i = f_{\theta_2}(x_i)$. Here two MLP layers $f_{\theta_1}$ and $f_{\theta_2}$ are used to encode the mean $u_i$ and standard deviation $\sigma_i$ of $z_i$, and another MLP layer $f_{\theta_3}$ is used to map the $z_i$ into the logits $\hat{y}_i^{ib}$ using $\hat{y}_i^{ib} = f_{\theta_3}(z_i)$. And the re-parameterization trick [37] is also adapted as: $z_i = \mu_i + \epsilon \sigma_i, \epsilon \in \mathcal{N}(0, \mathbf{I})$. Here $z_i$ is the compressed denoised representation of $x_i$, and the whole representation of each modality can be denoted as $F_m \in \mathbb{R}^d$. Through assigning $X = R_m$ and $Z = F_m$ in Eq. 3, we obtain the compressed hyper-modality enhanced representations $F_m$ and their corresponding standard deviation $\sigma_m$ for each modality $m$.

## 3.4 Multimodal Hyper-Modality Fusion Module

The effectiveness of multimodal fusion depends on the reliability of the representations from each modality. As the representation power of compressed representations can vary, it is essential to account for this variability during the fusion process. Drawing inspiration from the EAU framework [36], which uses variance as a measure of uncertainty in representations that higher variance indicates more scattered representations, we incorporate the standard deviation of hyper-modality representations to guide their fusion with the original representations.

Specifically, we compute the reciprocal of the standard deviation, expressed as $\zeta_m = \frac{1}{\sigma_m + eps}$, where $eps = 10^{-8}$ prevents division by zero. Additionally, since the difference of the standard deviation may very high, we limit the influence of uncertainties using thresholds $t_l$ and $t_u$, constraining $\zeta_m$ to lie within $[t_l, t_u]$. Besides, the contribution of the enhanced hyper-modality should not exceed the original modality representation, thus we define the uncertainty weights as: $U_m^2 = [1.0, \zeta_m]$ to fuse the $H_m$ and $F_m$. Furthermore, the interactions between the hyper-modality enhanced representations $F_m$ may also provide sentiment-relevant information, thus we define the uncertain weights of these three representations as:

$$U_m^3 = \frac{e^{\zeta_m}}{\sum_{m \in \{L, V, A\}} e^{\zeta_m}}. \tag{5}$$

Then the uncertainty weights are combined with attention weights [30] for further fusion through:

$$A_m^2 = \frac{[H_m, F_m]W_{a,m}W_{h,m}^\top}{\sqrt{d}},\qquad(6)$$

and

$$A_m^3 = \frac{F_{lav}W_a W_{a,h}^\top}{\sqrt{d}},\qquad(7)$$

where $W_{a,m}$, $W_{h,m}^\top$, $W_a$, $W_{a,h}^\top$ are trainable parameters, and $F_{lav} = [F_L, F_V, F_A]$.

The integrated representation of each modality is then fused with $A_m^2$ through: $S_m = Softmax(U_m^2 A_m^2)[H_m, F_m]$, and the fused representations of three hyper-modality enhanced representations is calculated as: $S_H = Softmax(U_m^3 A_m^3)F_{lav}$. The fused representations are then concatenated across modalities to form $S_{\text{all}} = [S_L, S_V, S_A, S_H]$, where $S_{\text{all}} \in \mathbb{R}^{4 \times d}$. $S_{all}$ is then fed into a transformer encoder, denoted as $\mathcal{F}_\sigma(\cdot)$ for further fusion. And the output, $H_{\text{final}}$, is flattened into $\mathbb{R}^{4d}$ and passed through a two-layer MLP $f_{\theta_4}$. The MLP reduces the representation first to $\mathbb{R}^{2d}$ and then to $\mathbb{R}^k$, producing the final prediction $\hat{y}$. Here, $k$ represents the dimensionality of the task labels. The process can be formulated as: $\hat{y} = f_{\theta_4}(\mathcal{F}_\sigma(S_{all}))$.

**Overall Loss.** The overall loss for the HME framework is defined as:

$$\mathcal{L}_{all} = \mathcal{L}_{TASK}(\hat{y}, y) + \alpha \mathcal{L}_{VIB}.\qquad(8)$$

Here the $\mathcal{L}_{VIB}$ is averaged over the three modalities, and $\mathcal{L}_{TASK}$ for regression and classification tasks is defined as:

$$\mathcal{L}_{TASK} = \begin{cases} CrossEntropy(\hat{y}, y) & Classification \\ |\hat{y} - y| & Regression \end{cases}.\qquad(9)$$

## 4 Experiments

### 4.1 Datasets and Implementation Details

**Datasets.** Three widely-used datasets are adapted, named CMU-MOSI [38], CMU-MOSEI [39] and IEMOCAP [40]. CMU-MOSI and CMU-MOSEI consist of 2,199 and 22,856 video clips, respectively. Each clip has a regression label ranging from -3 (strongly negative) to 3 (strongly positive). While IEMOCAP is a dialogue dataset with 4,453 samples, categorized into four emotional classes: neutral, happy, sad, and angry. Following prior research [29, 13], the 'positive/negative' accuracy and F1-score is adapted in MOSI and MOSEI, and averaged accuracy and F1-score of four classes is used to evaluate the IEMOCAP.

**Feature Extraction.** For text, the pre-trained BERT-base model [41] is used to obtain contextual embeddings. For the video modality, we perform face detection and feature extraction using the MTCNN [42] and OpenFace [43] toolkits. For audio, Mel-frequency cepstral coefficients and pitch features are extracted using the COVAREP toolkit [44].

**Implementation Details.** The performance of each model is evaluated with two protocols, fixed missing protocol and random missing protocol. For the former protocol, we test the performance on the test set with the same missing modality. For example, 'L' in Table 1 denotes that only language modality is available. Under this protocol, the modality is only removed during validation and inference—not during training, which aligns with following existing approaches [13, 15, 24, 29] and is intended to simulate real-world scenarios where certain modalities may be missing only at test time. While for random missing protocol, following existing approaches [13, 15, 24], we randomly drop some modalities for each sample. Here missing rate (MR) is adapted to evaluate the missingness of the dataset. The MR is defined as $MR = 1 - \frac{\sum_{i=1}^{N} a_i}{3 \times N}$, where $a_i$ is the number of the available modalities in $i^{th}$ sample, 'N' denotes the total number of the samples in each dataset. Since at least one modality is present, so the $a_i \geq 0$ and $MR \leq \frac{2}{3}$. Following GCNet [13], we also choose the MR from $[0.0, 0.1, ..., 0.7]$, where $0.7$ is the approximation of $\frac{2}{3}$. The MR is kept the same in both training, validation and testing phase. To mimic the real-world situations, we set the paradigm in training, validation and testing phase that the modalities are dropped and can not be used for guidance for reconstruction. Due to the space limits, more implementation details can be found in Appendix A.2.

Table 1: Performance on MOSI and IEMOCAP with missing modalities. 'ACC/F1' is reported. '+': results under fixed missing protocol; '-': results under random missing protocol.

| Datasets | Avail | GCNet | MPLMM | DiCMoR | IMDer | LNLN | HME |
|---|---|---|---|---|---|---|---|
| MOSI$^+$ | L | 83.7/83.6 | 83.8/83.8$^\dagger$ | 84.5/84.4 | 84.8/84.7 | 84.9/85.1 | **85.7/85.6** |
| | V | 56.1/55.7 | 54.7/42.1$^\dagger$ | 62.2/60.2 | 61.3/60.8 | 52.2/58.9 | **63.6/63.4** |
| | A | 56.1/54.5 | 54.6/42.0$^\dagger$ | 60.5/60.8 | 62.0/**62.2** | 52.2/58.9 | **62.7**/62.2 |
| | L,V | 84.3/84.2 | 83.8/83.7$^\dagger$ | 85.5/85.4 | 85.5/85.4 | 84.9/85.1 | **85.8/85.8** |
| | L,A | 84.5/84.4 | 83.7/83.6$^\dagger$ | 85.5/85.5 | 85.4/85.3 | 84.9/85.2 | **85.8/85.7** |
| | A,V | 62.0/61.9 | 57.3/42.6$^\dagger$ | 64.0/63.5 | 63.6/63.4 | 52.2/58.9 | **64.6/64.5** |
| | L,A,V | 85.2/85.1 | 83.8/83.8$^\dagger$ | 85.7/85.6 | 85.7/85.6 | 84.3/84.6 | **86.4/86.3** |
| | Avg. | 73.1/72.8 | 71.7/65.9$^\dagger$ | 75.4/75.1 | 75.5/75.3 | 70.8/73.8 | **76.4/76.2** |
| IEMOCAP$^+$ | L | -/76.1 | 69.2/69.3 | 75.2/68.2$^\dagger$ | 75.1/67.2$^\dagger$ | 76.8/74.4$^\dagger$ | **79.1/78.1** |
| | V | -/61.6 | 57.6/57.0 | 73.6/68.8$^\dagger$ | 75.1/70.1$^\dagger$ | 74.5/69.9$^\dagger$ | **75.7/70.8** |
| | A | -/63.5 | 59.8/59.7 | 76.0/68.7$^\dagger$ | 75.5/70.2$^\dagger$ | 77.1/74.6$^\dagger$ | **78.9/76.8** |
| | L,V | -/77.4 | 74.7/74.5 | 74.4/72.1$^\dagger$ | 75.2/72.6$^\dagger$ | 77.0/75.5$^\dagger$ | **79.7/78.9** |
| | L,A | -/79.1 | 76.0/75.4 | 78.7/76.5$^\dagger$ | 76.6/71.7$^\dagger$ | 77.9/75.4$^\dagger$ | **80.7/80.0** |
| | A,V | -/65.4 | 67.3/67.4 | 76.3/75.2$^\dagger$ | 76.3/72.3$^\dagger$ | 77.3/74.9$^\dagger$ | **78.1/76.3** |
| | L,A,V | -/**82.7** | -/- | 78.7/76.5$^\dagger$ | 77.5/75.4$^\dagger$ | 78.4/77.4$^\dagger$ | **81.1**/80.6 |
| | Avg. | -/72.3 | 67.4/67.2 | 76.1/72.3$^\dagger$ | 75.9/71.4$^\dagger$ | 77.0/74.6$^\dagger$ | **79.0/77.4** |
| MOSI$^-$ | 0.0 | 85.2/85.1 | 83.8/83.8$^\dagger$ | 85.7/85.6$^\dagger$ | 85.7/85.7$^\dagger$ | 84.2/84.0$^\dagger$ | **86.4/86.3** |
| | 0.1 | 82.3/82.3 | 80.6/80.7$^\dagger$ | 83.2/83.2$^\dagger$ | 84.6/84.4$^\dagger$ | 81.4/81.4$^\dagger$ | **84.9/84.7** |
| | 0.2 | 79.4/79.5 | 77.7/77.9$^\dagger$ | 81.6/81.3$^\dagger$ | 81.7/81.8$^\dagger$ | 78.7/78.7$^\dagger$ | **82.9/82.9** |
| | 0.3 | 77.2/77.2 | 75.0/75.1$^\dagger$ | 78.4/77.9$^\dagger$ | 79.9/79.4$^\dagger$ | 74.7/74.8$^\dagger$ | **81.1/81.0** |
| | 0.4 | 74.3/74.4 | 70.0/69.8$^\dagger$ | 76.2/74.7$^\dagger$ | 78.1/77.2$^\dagger$ | 70.3/69.9$^\dagger$ | **79.9/80.0** |
| | 0.5 | 70.0/69.8 | 67.4/66.6$^\dagger$ | 72.6/72.7$^\dagger$ | 73.6/73.7$^\dagger$ | 67.1/66.3$^\dagger$ | **76.4/76.4** |
| | 0.6 | 67.7/66.7 | 63.4/61.8$^\dagger$ | 71.3/71.4$^\dagger$ | 72.9/69.8$^\dagger$ | 62.5/60.5$^\dagger$ | **74.5/74.4** |
| | 0.7 | 65.7/65.4 | 59.2/56.2$^\dagger$ | 69.1/69.2$^\dagger$ | 67.7/67.1$^\dagger$ | 61.0/58.3$^\dagger$ | **73.5/71.7** |
| | Avg. | 75.2/75.1 | 72.1/71.5$^\dagger$ | 77.3/77.0$^\dagger$ | 78.0/77.4$^\dagger$ | 72.5/71.7$^\dagger$ | **80.0/79.7** |
| IEMOCAP$^-$ | 0.0 | -/78.4 | 76.4/72.0$^\dagger$ | 78.7/76.5$^\dagger$ | 77.5/75.4$^\dagger$ | 78.4/77.4$^\dagger$ | **81.1/80.6** |
| | 0.1 | -/77.5 | 76.1/70.8$^\dagger$ | 77.1/73.7$^\dagger$ | 77.2/72.8$^\dagger$ | 78.0/76.3$^\dagger$ | **80.9/80.3** |
| | 0.2 | -/77.3 | 75.7/67.9$^\dagger$ | 76.4/72.5$^\dagger$ | 75.9/71.6$^\dagger$ | 77.6/75.7$^\dagger$ | **80.7/80.0** |
| | 0.3 | -/76.2 | 75.6/67.4$^\dagger$ | 75.8/71.6$^\dagger$ | 75.7/70.8$^\dagger$ | 77.5/75.4$^\dagger$ | **79.0/77.8** |
| | 0.4 | -/75.1 | 75.5/66.6$^\dagger$ | 75.5/69.0$^\dagger$ | 74.9/70.8$^\dagger$ | 77.4/74.0$^\dagger$ | **78.7/77.3** |
| | 0.5 | -/73.8 | 74.0/66.2$^\dagger$ | 75.1/68.5$^\dagger$ | 74.7/68.2$^\dagger$ | 76.8/72.9$^\dagger$ | **78.5/77.1** |
| | 0.6 | -/71.9 | 73.2/66.0$^\dagger$ | 74.5/68.0$^\dagger$ | 73.5/67.0$^\dagger$ | 76.6/72.6$^\dagger$ | **77.8/76.1** |
| | 0.7 | -/71.4 | 72.4/65.7$^\dagger$ | 74.3/67.5$^\dagger$ | 73.1/65.3$^\dagger$ | 76.5/70.2$^\dagger$ | **77.6/76.0** |
| | Avg. | -/75.2 | 74.9/67.8$^\dagger$ | 75.9/70.9$^\dagger$ | 75.3/70.2$^\dagger$ | 77.4/74.3$^\dagger$ | **79.3/78.2** |

## 4.2 Comparison with the State-of-the-arts

We compare HME with the state-of-the-art models specifically designed for MSA under missing modality scenarios, including GCNet [13], IMDer [15], DiCMoR [24], LNLN [14], and MPLMM [29]. In some cases, we were unable to directly compare with results reported in prior work under identical settings—either because the specific missing modality protocol was not used, or the results for certain datasets were not reported for those conditions. For a fair comparison, we re-implement the models from their open-source codes according to our training paradigm, which are denoted with symbol $^\dagger$. More results on MOSEI dataset can be found in Appendix A.3.1.

**Quantitative Results.** Table 1 summarizes the performance of HME and baselines across MOSI and IEMOCAP datasets under the both missing modality protocols. Under fixed missing protocol, HME generally outperforms existing models. On MOSI dataset, HME surpasses all baselines across every metric, achieving an average performance increase of 0.9. For the IEMOCAP dataset, HME performs comparably to GCNet when all modalities are available but demonstrates significant advantages over other models in scenarios with missing modalities. On average, across all missing modality scenarios, HME improves ACC by 2.0 and F1 score by 2.8 over the best-performing baseline. This strong performance extends to the random missing protocol. On both MOSI and IEMOCAP datasets, HME consistently outperforms all models across all metrics and at every MR. Specifically, HME yields

Table 2: Ablation results of HME components on MOSI and IEMOCAP datasets.

| Datasets | HME | w/o $E_L$ | w/o $E_V$ | w/o $E_A$ | w/o $E_S$ | w/o $U_m$ | w/o HMG |
|---|---|---|---|---|---|---|---|
| Ablations on Fixed Missing Protocol | | | | | | | |
| MOSI | **76.4/76.2** | 75.2/74.7 | 75.2/75.1 | 75.3/74.6 | 75.2/74.7 | 75.4/75.1 | 75.0/74.5 |
| IEMOCAP | **79.0/77.4** | 78.4/76.3 | 78.1/76.2 | 78.5/76.5 | 78.2/76.1 | 79.1/76.3 | 78.5/76.8 |
| Ablations on Random Missing Protocol | | | | | | | |
| MOSI | **79.0/78.7** | 77.3/76.8 | 77.3/76.7 | 76.7/76.2 | 77.2/76.9 | 77.6/77.2 | 76.6/76.2 |
| IEMOCAP | **79.0/77.8** | 78.4/76.8 | 78.4/76.7 | 78.3/76.5 | 78.4/76.9 | 78.2/76.4 | 78.7/77.2 |

average improvements in ACC and F1-score of 2.0 and 2.3, respectively, on MOSI, and 1.9 and 3.9, respectively, on IEMOCAP. The consistent superiority of HME across both missing protocols underscores its effectiveness and robustness in handling various missing modalities.

## 4.3 Ablation Study

To better understand the contribution of each component in HME, we conduct an ablation study on the following parts across both missing modality protocols: (i) 'w/o $E_m$' refers to the model without the hyper-modality enhanced representation of modality $m$; (ii) 'w/o $E_S$' refers to the model without the overall representation of the current sample; (iii) 'w/o $U_m$' refers to the model without uncertainty weights, using only vanilla attention weights for fusion; and (iv) 'w/o $HMG$' refers to the model without the hyper-modality generation module, where the modality representations $E_m$ are directly passed into the VIB for further processing. The results of averaged performance of two missing protocols are presented in Table 2, with more detailed results in Appendix A.3.2.

**Effects of the HME components.** In both missing protocols, removing any component resulted in a decline in performance across all datasets and missing protocols, highlighting the effectiveness of each part of HME. Moreover, the importance of specific components varies across datasets. For example, removing the HMG module causes the largest performance drop in MOSI datasets under both missing protocols, whereas on the IEMOCAP dataset, the removal of $U_m$ results in the largest performance decline. Furthermore, the removal of uncertainty weights $U_m$ also leads to a substantial drop in performance, particularly in F1 score, emphasizing the role of uncertainty in accurately guiding the model to distinguish between sentiment categories. Overall, $U_m$ and HMG contribute the most to performance improvement. For example, removing HMG leads to the largest performance drop on the MOSI and MOSEI datasets under the fixed missing protocol, while removing $U_m$ has the most significant impact under the random missing protocol in IEMOCAP and MOSEI.

## 4.4 Visualization of the Training Loss and Representations

To better analyze the convergence of the loss components during training, we conduct experiments on the IEMOCAP dataset with an MR value of 0.7, which is depicted in Figure 3 (a). Note that the $L_{VIB,1}$ and $L_{VIB,2}$ are the two components of $L_{VIB}$. Specifically, training is conducted over 100 epochs, with early stopping applies at the 60th epoch. We observe that all loss components of HME gradually decrease over time and stabilize after a few epochs, indicating the rational design of HME's loss components. This consistent reduction in both overall and local losses, with minimal fluctuations, further validates the effectiveness of HME's loss component structure.

Besides, to understand the differences between the enhanced hyper-modality representations $F_m$ and the original representations $H_m$, we visualize them using t-SNE. Figure 3 (b) shows these visualizations on the MOSI test set with an MR of 0.7. We find that the enhanced hyper-modality representation $F_m$ captures additional sentiment-relevant information, forming distinct clusters that do not overlap with the original representations $H_m$. More visualizations for the fused representations $S_m$ and $S_H$ can be found in Appendix A.3.9.

## 4.5 Qualitative Analysis

To investigate the relationship between uncertainty weights and the input modalities, we compute the average uncertainty scores across three modalities under 14 missing-modality scenarios derived from two missing-protocol setups on the validation set. Each scenario is executed five times to ensure reliability, and the results, including 95% confidence intervals, are presented in Table 3. Further

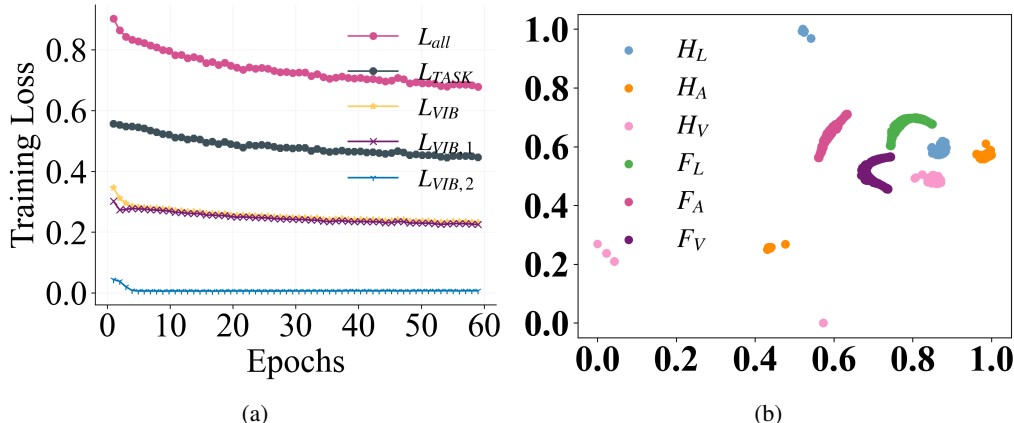

(a)  (b)

Figure 3: Visualization of training loss convergence on the IEMOCAP dataset (a) and learned representations on the MOSI dataset (b) with $MR = 0.7$. (a) All losses show clear convergence during training. (b) The enhanced hyper-modality representations capture distinct information compared to the original ones, forming separate clusters.

Table 3: Average uncertainty weights of the three modalities on IEMOCAP dataset.

| MR | 0.1 | 0.2 | 0.3 | 0.4 | 0.5 | 0.6 |
|---|---|---|---|---|---|---|
| $F_L$ | 0.426±0.023 | 0.424±0.011 | 0.341±0.007 | 0.336±0.027 | 0.323±0.019 | 0.322±0.033 |
| $F_A$ | 0.244±0.034 | 0.248±0.008 | 0.299±0.029 | 0.319±0.046 | 0.338±0.037 | 0.348±0.033 |
| $F_V$ | 0.330±0.034 | 0.328±0.004 | 0.360±0.025 | 0.345±0.022 | 0.339±0.031 | 0.330±0.029 |
| | L | A | V | L,A | L,V | A,V |
| $F_L$ | 0.290±0.034 | 0.273±0.054 | 0.322±0.001 | 0.301±0.007 | 0.312±0.028 | 0.364±0.015 |
| $F_A$ | 0.350±0.049 | 0.467±0.134 | 0.389±0.005 | 0.355±0.013 | 0.416±0.062 | 0.311±0.026 |
| $F_V$ | 0.360±0.028 | 0.260±0.104 | 0.289±0.004 | 0.344±0.012 | 0.272±0.041 | 0.325±0.017 |
| $t_s$ | 0.1 | 0.2 | 0.3 | 0.4 | L,A,V($t_s$0.7) | MR=0.7 |
| $F_L$ | 0.358±0.012 | 0.353±0.016 | 0.432±0.003 | 0.461±0.003 | 0.409±0.003 | 0.293±0.026 |
| $F_A$ | 0.219±0.013 | 0.238±0.017 | 0.247±0.004 | 0.230±0.003 | 0.257±0.009 | 0.365±0.021 |
| $F_V$ | 0.423±0.010 | 0.403±0.012 | 0.318±0.004 | 0.310±0.003 | 0.335±0.005 | 0.342±0.031 |

analyses of model stability (in A.3.3), error (in A.3.4), generalization (in A.3.5), VIB components (in A.3.6), enhanced representations (in A.3.7), hyper-parameters (in A.3.8), case visualization (A.3.9) and computational efficiency (in A.3.10) are detailed in the Appendix.

Our findings reveal that the uncertainty weights of the three modalities are strongly influenced by the number of missing modalities (i.e., the MR value). Specifically: (i) When fewer modalities are missing (low MR, e.g., MR = 0.0, 0.1, 0.2): The language modality exhibits the smallest variance in its enhanced representations, providing the most information, while the audio modality shows the largest variance. This may explain why prior studies often regard language as the primary modality in fusion process [14, 21]. (ii) As the MR value increases: The audio modality's variance gradually decreases, and its uncertainty weights increase, reflecting its growing contribution to the fusion process. Conversely, the language modality's uncertainty weights decrease as its representation variance grows. (iii) Similar trends are observed under the fixed missing protocol. For example, when only one modality is present ($MR = 0.7$), the audio modality tends to maintain smaller variance.

Since the hyper-modality representations are generated from representations that satisfy the similarity threshold $t_s$, we further analyze the relationship between uncertainty weights and $t_s$. Specifically, we conduct experiments under the full-modality scenario ($MR = 0.0$, corresponding to L,A,V) to isolate the impact of $t_s$ on uncertainty weights, which is shown in Table 3. Our findings reveal the following: (i) When $t_s$ is low (e.g., 0.1 or 0.2): The video modality demonstrates the smallest variance in its representations, resulting in the highest uncertainty weights, while the audio modality exhibits the largest variance. (ii) As $t_s$ increases, The language modality's uncertainty weights rise, gradually aligning with its behavior in low-MR scenarios, where it dominates with higher uncertainty

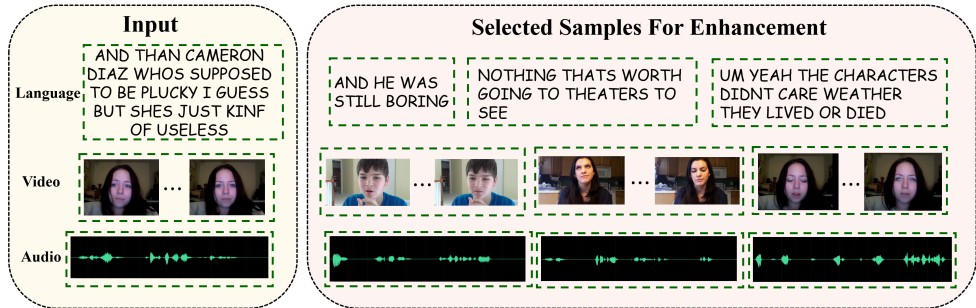

Figure 4: Case visualization of selected samples with $t_s = 0.9$ on the MOSI dataset. Green boxes indicate representations with the same label, while red boxes denote samples with different labels.

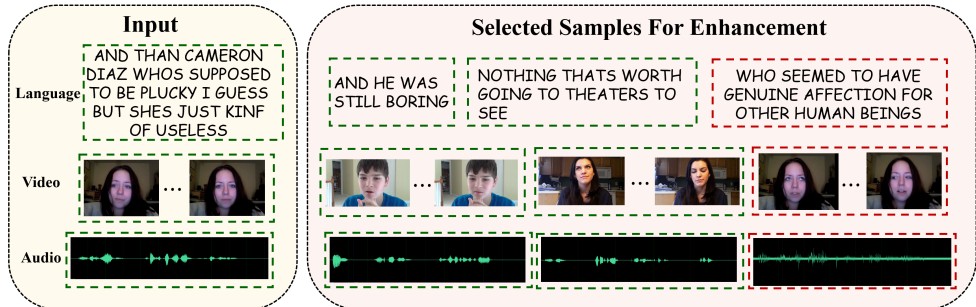

Figure 5: Case visualization of selected samples with $t_s = 0.5$ on the MOSI dataset. Green boxes indicate representations with the same label, while red boxes denote samples with different labels.

weights. (iii) Across all $t_s$ values: The audio modality's uncertainty weights remain consistently low, mirroring its behavior in low-MR scenarios.

To better understand the the proposed modality enhancement, we visualize selected sample pairs from the MOSI dataset under different similarity thresholds. Specifically, we examine two configurations with a MR of 0.7 and a batch size of 128: (i) a similarity threshold of 0.9 (Figure 4), and (ii) a similarity threshold of 0.5 (Figure 5). A higher similarity threshold tends to select samples that are more semantically consistent with the current label. Conversely, lowering the threshold increases sample diversity but also introduces a few mismatched labels. Nonetheless, the subsequent VIB and uncertainty-aware fusion mechanism effectively reduce the impact of these mismatches. Further analysis of the effects of batch size and selected samples is provided in the Appendix A.3.9.

## 5 Conclusion and Discussion

In this paper, we present the Hyper-Modality Enhancement (HME) framework to address the challenges posed by missing modalities in multimodal sentiment analysis. By integrating sentiment-relevant information from other samples, HME effectively enhances the representation of the sample. The uncertainty-aware fusion strategy further ensures that the enhanced representations are robust and reliable, even when generated data varies in quality. Experimental evaluations on multiple benchmark datasets demonstrate that HME outperforms existing state-of-the-art methods, achieving superior performance in terms of both accuracy and robustness when handling missing modalities.

**Discussion of Limitation.** While the use of VIB helps reduce redundant information, generating separate representations for each modality can still result in overlapping content. Besides, HME only focuses on mitigating noise from other samples but does not account for potential noise within the original modality representations themselves. We plan to address these challenges in future work.

**Discussion of Societal Impacts.** Recognizing emotions from incomplete multimodal data enhances the robustness and usability of MSA systems but also raises ethical concerns. Inference from partial signals may lead to inaccurate or biased predictions, potentially misrepresenting individuals or groups. Moreover, the use of MSA models can introduce privacy risks or enable surveillance without consent.

## Acknowledgments

This work was supported by the National Natural Science Foundation of China (Grant No.U24A20250 and No.62176048), the Natural Science Foundation of Sichuan Province (Grant No.2024NSFTD0042, No.2024YFG0006, No.2024NSFSC0506, and No.2025ZNSFSC1487), and the Fundamental Research Funds for the Central Universities (No.ZYGX2024J022 and No.ZYGX2024Z005).

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

# A Technical Appendices and Supplementary Material

## A.1 Additional HME Details

In this section, we provide further details of the HME framework. After the sample selection step in Eq.1, the extracted features $H'_m$ are passed into a Perceiver network [32] to generate enhanced representations. The Perceiver refines the features through a combination of cross-modal and self-attention mechanisms [30], enabling richer information exchange across modalities. The Perceiver operates by introducing a set of learnable latent vectors, denoted as $E'_m \in \mathbb{R}^{l_p \times d}$, which interact with the input features $H'_m$. Each Perceiver layer alternates between two key components: a cross-attention block and a self-attention block [32]. In the cross-attention block, the latent vectors $E'_m$ act as the queries, while the input features $H'_m$ serve as the keys and values. This process allows the latent vectors to selectively attend to informative aspects of $H'_m$, formulated as:

$$H_{ca} = ATTN_{\phi_{m,m}}(E'_m, H'_m) = \text{Softmax}(\frac{E'_m W_{Q_e} W_h^\top (H'_m)^\top}{\sqrt{d}})H'_m W_{V_h}, \tag{10}$$

where $W_{Q_e}, W_h, W_{V_h} \in \mathcal{R}^{d \times d}$ are trainable weights, and $H_{ca} \in \mathcal{R}^{l_p \times d}$. The resulting features are then normalized and refined through a residual connection:

$$\hat{H}_{ca} = LN(E'_m, H_{ca}) + E'_m, \tag{11}$$

followed by a feed-forward update:

$$H_{ff} = LN(W_{ff}\hat{H}_{ca}) + \hat{H}_{ca}, \tag{12}$$

where $W_{ff}$ is trainable. The subsequent self-attention block applies the same mechanism but sets the queries, keys, and values all to the same input (e.g., $H_{ff}$). By stacking multiple layers of these alternating blocks, the Perceiver gradually produces a enhanced representation $E'_m$. Finally, the enhanced modality representation $E_m \in \mathbb{R}^d$ is obtained by averaging the outputs of the last transformer layer.

## A.2 Additional Implementation Details

In this section, we describe the datasets, and implementation details used in experiments.

**MUStARD and UR-FUNNY datasets.** We conduct additional experiments on the UR-FUNNY [45] and MUStARD [46] datasets, both designed for humor detection from multimodal data. The UR-FUNNY dataset contains 1,866 videos from 1,741 speakers, comprising 9,588 utterances. The data are divided into 7,614 training, 980 validation, and 994 test instances. The MUStARD dataset includes 690 videos, split into 539 training, 68 validation, and 68 test utterances.

**Implementation Details.** To ensure reliable training and prevent overfitting or underfitting, we train 100 epochs and apply early stopping with a patience of 10 epochs across all reproduced baseline models and HME. Specifically, training is terminated when the validation loss fails to improve for 10 consecutive epochs. Besides, for both missing modality protocols, the reported results for HME and re-implemented baselines (denoted with symbol $^\dagger$) are the averages from five separate runs. All models are evaluated on an NVIDIA GTX 3090 GPU.

For HME under the two missing protocols across three datasets, several hyper-parameters remain consistent, such as the Perceiver layers in the modality enhancement module, which are set to 3 layers, the prompt length $l_p = 5$, the similarity threshold $t_s = 0.6$, and the bounds of uncertainty weights $t_l = 0.2$, $t_u = 1.0$. However, some hyper-parameters vary depending on the specific dataset and configuration. For the MOSI and MOSEI datasets, the default hyper-parameters under both settings are as follows: learning rate of 4e-5, batch size of 256, and VIB loss weight $\alpha = 1.0$. The hidden dimension $d$ for MOSI and MOSEI also differ across protocols. For MOSI, the fixed missing protocol uses a hidden dimension $d$ of 96, while the random missing protocol uses 192. In the case of MOSEI, the hidden dimension $d$ is set to 192 for both protocols. In contrast, for the IEMOCAP dataset, the default hyper-parameters are consistent across both missing protocols, with a learning rate of 1e-4, batch size of 24, VIB loss weight $\alpha = 0.5$, and hidden dimension $d = 30$.

To identify the optimal hyper-parameters of HME, we performed a search over a limited set of values. For the MOSI and MOSEI datasets, the learning rate was selected from {1e-5, 2e-5, 3e-5, 4e-5}, the

Table 4: Performance comparison under fixed missing protocol on MOSEI.

| Datasets | Avail | GCNet | MPLMM[†] | DiCMoR | IMDer | LNLN | HME |
|----------|-------|-------|----------|--------|-------|------|-----|
| MOSEI | L | 83.0/83.2 | 84.6/84.6 | 84.2/84.3 | 84.5/84.5 | 84.1/84.4 | **85.5/85.4** |
| | V | 61.9/61.6 | 62.2/53.2 | 63.6/63.6 | 63.9/63.6 | 62.9/**77.2** | **64.6**/62.9 |
| | A | 60.2/60.3 | 61.8/49.4 | 62.9/60.4 | 63.8/60.6 | 62.9/**77.2** | **64.1**/59.7 |
| | L,V | 84.3/84.4 | 85.2/85.1 | 84.9/84.9 | 85.0/85.0 | 84.4/84.7 | **85.7/85.7** |
| | L,A | 84.3/84.4 | 85.1/85.1 | 85.0/84.9 | 85.1/85.1 | 84.1/84.4 | **85.6/85.4** |
| | A,V | 64.1/57.2 | 63.2/51.0 | 65.2/64.4 | 64.9/63.5 | 62.9/**77.2** | **65.4**/63.7 |
| | L,A,V | 85.2/85.1 | 85.3/85.3 | 85.1/85.1 | 85.1/85.1 | 84.1/84.5 | **86.2/86.2** |
| | Avg. | 74.7/73.7 | 75.3/70.5 | 75.8/75.4 | 76.0/75.3 | 75.1/**81.4** | **76.7**/75.6 |

Table 5: Performance comparison under random missing protocol on MOSEI.

| Datasets | MR | GCNet | MPLMM[†] | DiCMoR[†] | IMDer[†] | LNLN[†] | HME |
|----------|-----|-------|----------|-----------|----------|---------|-----|
| MOSEI | 0.0 | 85.2/85.1 | 85.3/85.3 | 85.1/85.1 | 85.1/85.1 | 85.1/85.1 | **86.2/86.2** |
| | 0.1 | 82.3/82.1 | 83.7/83.5 | 83.2/83.1 | 83.5/83.3 | 83.1/83.0 | **84.0/83.8** |
| | 0.2 | 80.3/79.9 | 81.4/80.8 | 81.2/80.8 | 81.3/80.9 | 81.7/81.0 | **82.3/81.9** |
| | 0.3 | 77.5/76.8 | 79.2/78.3 | 79.0/77.9 | 79.4/78.8 | 79.1/78.1 | **80.0/79.6** |
| | 0.4 | 76.0/74.9 | 77.1/76.1 | 76.9/75.2 | 76.1/74.3 | 76.9/75.6 | **77.6/76.4** |
| | 0.5 | 74.9/73.2 | 75.6/73.8 | 73.7/71.7 | 75.3/72.4 | 75.1/72.4 | **75.7/74.7** |
| | 0.6 | **74.1**/72.1 | 72.7/70.6 | 71.1/70.4 | 71.2/66.5 | 72.5/68.6 | 73.0/**72.5** |
| | 0.7 | **73.2/70.4** | 70.5/69.2 | 70.6/68.9 | 71.1/65.9 | 70.6/65.2 | 72.4/69.2 |
| | Avg. | 77.9/76.8 | 78.2/77.2 | 77.6/76.6 | 77.9/75.9 | 78.0/76.1 | **78.9/78.0** |

VIB loss weight $\alpha$ from {0.5, 0.6, 0.7, 0.8, 0.9, 1.0}, and the hidden dimension $d$ from {92, 128, 160, 192, 256}. For the IEMOCAP dataset, the learning rate was chosen from {1e-4, 5e-4, 1e-3, 2e-3}, the VIB loss weight $\alpha$ from {0.5, 0.6, 0.7, 0.8, 0.9, 1.0}, and the hidden dimension $d$ from {30, 35, 40, 45, 50, 55}.

To identify the hyper-parameter of the re-implemented baselines, we adhered to the hyper-parameter guidelines provided in the official implementations of these models. For models like IMDer [15] and DiCMoR [24], which included pre-trained weights, we used the publicly available checkpoints and configurations. For baselines without available hyper-parameters for specific datasets, we aligned their hyper-parameters with HME's settings where possible. When unique parameters were required, we referred to their recommended settings from other publicly available implementations.

Since MOSI and MOSEI are regression tasks, consistent with previous work [29], we convert the ground truth and predictions into two categories: greater than 0 and less than 0. This enables the calculation of binary classification metrics such as accuracy and F1-score.

### A.3 Additional Results

Here we present additional experimental results in this section. These include the performance of different models on MOSEI dataset (in A.3.1), the ablation studies for each component under various missing modality scenarios across three datasets (in A.3.2), model stability analysis (in A.3.3), error analysis (in A.3.4), generalization and plug-and-play applicability analysis (in A.3.5), VIB components analysis (in A.3.6), enhanced representations analysis (in A.3.7), hyper-parameter analysis(in A.3.8), case visualization and representation visualization (in A.3.9), and computation overhead and trade-off analysis (in A.3.10).

### A.3.1 Results on MOSEI Dataset

In this section, we compare the performance of HME with other baselines on the MOSEI dataset under two missing protocols. The results are shown in Tables 4 and 5. Under the fixed missing protocol, HME performs slightly worse than LNLN in the missing language modality scenario in terms of F1 score, but it delivers strong results across other scenarios for both metrics. In the random missing protocol, HME performs slightly worse than GCNet at MR values of 0.6 and 0.7, but it still outperforms the current state-of-the-art models at all other missing rates. On average, HME improves

Table 6: Ablation results of fixed missing protocol on three datasets.

| Methods | L | V | A | L,V | L,A | A,V | L,A,V |
|---------|---|---|---|-----|-----|-----|-------|
| Ablations on MOSI | | | | | | | |
| HME | **85.7/85.6** | **63.6/63.4** | **62.7/62.2** | **85.8/85.8** | **85.8/85.7** | **64.6/64.5** | **86.4/86.3** |
| w/o $E_L$ | 85.4/85.0 | 61.7/60.3 | 61.1/60.2 | 85.7/85.4 | 84.6/84.2 | 62.5/62.6 | 85.4/85.2 |
| w/o $E_V$ | 85.1/84.8 | 61.3/61.2 | 62.0/62.1 | 84.9/84.6 | 85.7/85.4 | 62.5/62.3 | 85.2/85.2 |
| w/o $E_A$ | 84.9/84.9 | 60.5/58.1 | 62.3/61.0 | 85.2/85.1 | 85.5/85.3 | 63.0/61.8 | 85.8/85.7 |
| w/o $E_S$ | 85.4/85.0 | 61.7/60.3 | 61.1/60.2 | 85.7/85.4 | 84.6/84.2 | 62.5/62.6 | 85.5/85.4 |
| w/o $U_m$ | 85.2/85.0 | 61.0/60.8 | 61.1/60.7 | 85.7/85.6 | 85.7/85.4 | 62.7/61.9 | 86.1/86.0 |
| w/o HMG | 85.1/84.9 | 60.5/60.4 | 60.2/59.1 | 85.5/85.4 | 85.4/85.3 | 63.3/61.5 | 85.2/85.1 |
| w/o VIB | 84.8/84.6 | 61.1/61.3 | 61.9/61.6 | 85.7/85.6 | 85.2/85.0 | 62.7/62.5 | 85.4/85.2 |
| Ablations on MOSEI | | | | | | | |
| HME | **85.5/85.4** | **64.6/62.9** | **64.1/59.7** | **85.7/85.7** | **85.6/85.4** | **65.4/63.7** | **86.2/86.2** |
| w/o $E_L$ | 85.2/85.0 | 63.8/60.3 | 59.4/57.7 | 85.5/85.4 | 85.2/85.1 | 64.2/59.8 | 86.1/86.0 |
| w/o $E_V$ | 85.1/85.1 | 63.4/58.6 | 61.6/52.1 | 85.6/85.3 | 85.2/85.1 | 64.2/59.8 | 86.1/86.0 |
| w/o $E_A$ | 85.2/85.0 | 63.8/60.3 | 60.2/53.1 | 85.4/85.4 | 85.2/85.1 | 64.2/59.8 | 85.9/85.8 |
| w/o $E_S$ | 85.2/85.1 | 63.1/59.9 | 60.6/56.8 | 85.5/85.3 | 85.3/85.2 | 64.0/59.6 | 85.8/85.7 |
| w/o $U_m$ | 85.1/85.0 | 63.6/55.5 | 61.5/55.5 | 85.4/85.3 | 85.1/85.1 | 64.1/57.2 | 86.1/86.0 |
| w/o HMG | 85.3/85.2 | 62.8/55.5 | 61.0/51.9 | 85.5/85.3 | 85.1/85.0 | 63.5/53.1 | 85.6/85.6 |
| w/o VIB | 85.4/85.3 | 64.5/60.3 | 61.8/56.8 | 85.6/85.5 | 85.2/85.1 | 65.3/62.4 | 85.9/85.8 |
| Ablations on IEMOCAP | | | | | | | |
| HME | **79.1/78.1** | **75.7/70.8** | **78.9/76.8** | **79.7/78.9** | **80.7/80.0** | **78.1/76.3** | **81.1/80.6** |
| w/o $E_L$ | 78.8/77.6 | 74.3/68.0 | 78.0/75.5 | 79.5/78.3 | 80.4/79.4 | 77.2/75.0 | 80.8/80.0 |
| w/o $E_V$ | 78.2/77.4 | 74.1/70.0 | 78.1/75.7 | 79.1/78.0 | 80.5/79.4 | 75.8/72.5 | 80.9/80.2 |
| w/o $E_A$ | 78.7/77.9 | 74.5/69.2 | 78.6/75.9 | 79.2/78.3 | 80.4/79.3 | 77.5/74.5 | 80.9/80.4 |
| w/o $E_S$ | 78.2/77.4 | 74.0/69.0 | 78.8/75.6 | 78.9/78.1 | 80.2/79.2 | 77.1/74.3 | 80.0/79.4 |
| w/o $U_m$ | 78.2/77.5 | 73.9/69.0 | 78.3/75.4 | 78.4/77.9 | 80.3/79.5 | 76.7/74.4 | 81.0/80.2 |
| w/o HMG | 78.8/77.7 | 74.9/70.0 | 78.4/76.3 | 78.7/77.8 | 80.1/79.4 | 77.8/75.9 | 80.8/80.2 |
| w/o VIB | 78.9/77.7 | 73.9/69.9 | 78.4/75.9 | 79.1/78.1 | 80.3/79.6 | 76.7/74.1 | 80.6/79.7 |

the metrics by 0.7 and 0.8 across all eight MR values. These findings indicate that HME offers superior overall performance and greater robustness compared to existing state-of-the-art models.

### A.3.2 Detailed Ablation Results

In this section, we present detailed ablation results for both missing modality protocols, as shown in Tables 6 and 7. Due to space constraints in the main text, we add ablation experiments for the VIB component here, labeled as 'w/o VIB'. To maintain the integrity of the HMG component, the VIB ablation was performed by removing the $\mathcal{L}_{VIB}$ term in $\mathcal{L}_{all}$ during training.

Removing any component from HME consistently results in a performance drop across all missing modality scenarios. Specifically, removing the VIB leads to redundancy and interference from noisy information, reducing the model's effectiveness. Eliminating the enhanced hyper-modality representation $E_m$ results in a decline, likely because it removes sentiment-relevant information that could be provided by other samples for the current sample. Removing the $U_m$ component can result in lower-quality representations that are given more weight during the fusion process, especially in modalities with weaker representation capabilities, leading to a decrease in performance during fusion. Finally, removing the entire HMG module limits the model to using only individual modality representations from other samples, disregarding the interactions between these modalities, which are crucial for capturing sentiment-relevant information. These results highlight the critical importance and effectiveness of each component within the HME framework.

### A.3.3 Model Stability Analysis

Since modality missing occurs randomly, we conduct five separate runs to comprehensively evaluate the performance of HME and report the averaged results. To further assess the statistical significance of HME, we present the 95% confidence intervals for ACC and F1 scores under the random missing modality protocol on all three datasets, as shown in Table 8. Even in the worst-case scenario, HME consistently shows competitive performance, demonstrating its strong performance.

Table 7: Ablation results of random missing protocol on three datasets.

| Methods | 0.1 | 0.2 | 0.3 | 0.4 | 0.5 | 0.6 | 0.7 |
|---|---|---|---|---|---|---|---|
| | | | Ablations on MOSI | | | | |
| HME | **84.9/84.7** | **82.9/82.9** | **81.1/81.0** | **79.9/80.0** | **76.4/76.4** | **74.5/74.4** | **73.5/71.7** |
| w/o $E_L$ | 83.4/83.2 | 81.6/81.5 | 80.2/79.9 | 77.6/76.7 | 74.1/73.8 | 73.9/72.5 | 70.0/70.0 |
| w/o $E_V$ | 84.2/83.9 | 81.9/81.5 | 79.7/79.0 | 77.3/77.0 | 74.4/73.9 | 73.2/72.3 | 70.7/69.3 |
| w/o $E_A$ | 83.2/82.9 | 81.6/81.3 | 79.3/78.8 | 77.1/76.6 | 74.4/74.0 | 71.2/71.2 | 70.4/68.7 |
| w/o $E_S$ | 84.2/83.8 | 81.1/80.9 | 80.0/79.8 | 76.7/76.6 | 75.2/74.9 | 72.3/72.4 | 70.6/69.6 |
| w/o $U_m$ | 83.7/83.4 | 82.2/82.2 | 79.9/79.6 | 78.5/78.4 | 75.2/74.8 | 72.4/71.5 | 71.3/70.7 |
| w/o HMG | 83.7/83.7 | 80.8/80.8 | 80.0/79.3 | 75.3/75.2 | 74.1/74.1 | 72.4/71.2 | 69.8/69.2 |
| w/o VIB | 83.1/82.8 | 81.1/81.1 | 78.8/78.8 | 78.2/77.6 | 75.2/74.8 | 72.6/72.1 | 71.0/69.5 |
| | | | Ablations on MOSEI | | | | |
| HME | **84.0/83.8** | **82.3/81.9** | **80.0/79.6** | **77.6/76.4** | **75.7/74.7** | **73.0/72.5** | **72.4/69.2** |
| w/o $E_L$ | 83.8/83.6 | 81.8/81.1 | 79.7/78.8 | 77.1/75.9 | 75.2/73.9 | 72.5/71.2 | 71.0/66.4 |
| w/o $E_V$ | 83.7/83.4 | 81.6/81.0 | 79.0/78.6 | 76.8/75.6 | 74.8/73.5 | 71.8/70.9 | 70.9/65.8 |
| w/o $E_A$ | 83.5/83.2 | 81.7/81.1 | 79.3/78.6 | 77.1/75.2 | 74.8/73.1 | 72.6/69.5 | 70.8/65.8 |
| w/o $E_S$ | 83.7/83.4 | 81.7/81.2 | 79.2/78.0 | 77.1/75.3 | 75.3/73.6 | 72.3/71.0 | 71.5/68.7 |
| w/o $U_m$ | 83.2/83.0 | 81.5/80.7 | 79.5/78.0 | 76.6/75.0 | 74.5/71.6 | 72.5/70.1 | 70.8/65.6 |
| w/o HMG | 83.5/83.1 | 81.6/80.8 | 79.3/78.0 | 76.7/75.6 | 74.1/73.2 | 72.1/69.7 | 70.6/67.8 |
| w/o VIB | 83.7/83.3 | 81.7/81.2 | 79.1/78.6 | 76.7/76.0 | 75.2/73.9 | 72.7/71.0 | 71.0/67.7 |
| | | | Ablations on IEMOCAP | | | | |
| HME | **80.9/80.3** | **80.7/80.0** | **79.0/77.8** | **78.7/77.3** | **78.5/77.1** | **77.8/76.1** | **77.6/76.0** |
| w/o $E_L$ | 80.3/79.4 | 79.8/78.9 | 78.8/77.5 | 78.5/76.8 | 78.0/76.4 | 76.7/74.6 | 76.5/74.1 |
| w/o $E_V$ | 79.7/78.9 | 79.6/78.2 | 78.8/77.6 | 78.5/77.0 | 78.2/76.2 | 77.4/75.2 | 76.8/74.0 |
| w/o $E_A$ | 80.0/79.0 | 79.7/78.5 | 78.9/76.9 | 78.6/76.5 | 77.6/75.8 | 76.8/74.4 | 76.7/74.2 |
| w/o $E_S$ | 79.7/79.0 | 79.6/78.4 | 78.9/77.6 | 78.3/76.7 | 78.0/76.5 | 77.6/75.3 | 76.9/74.6 |
| w/o $U_m$ | 80.0/79.1 | 79.6/78.1 | 78.6/77.3 | 78.2/76.5 | 77.3/75.3 | 77.2/74.5 | 76.7/73.8 |
| w/o HMG | 80.3/79.6 | 80.0/79.0 | 78.9/77.3 | 78.5/77.0 | 78.2/76.4 | 77.7/75.7 | 77.3/75.4 |
| w/o VIB | 80.0/79.3 | 79.9/79.1 | 78.8/77.5 | 78.3/76.7 | 78.3/76.1 | 77.1/74.8 | 77.0/74.6 |

Table 8: Performance under random missing protocol on MOSI (95% confidence intervals).

| Datasets | 0.1 | 0.2 | 0.3 | 0.4 | 0.5 | 0.6 | 0.7 |
|---|---|---|---|---|---|---|---|
| | | | Statistical significance of ACC | | | | |
| MOSI | 84.9±0.2 | 82.9±0.2 | 81.1±0.2 | 79.9±0.7 | 76.4±0.3 | 74.5±0.6 | 73.5±0.9 |
| MOSEI | 84.0±0.2 | 82.3±0.3 | 80.0±0.3 | 77.6±0.2 | 75.7±0.3 | 73.0±0.2 | 72.4±0.5 |
| IEMOCAP | 80.9±0.2 | 80.7±0.3 | 79.0±0.4 | 78.7±0.3 | 78.5±0.3 | 77.8±0.2 | 77.6±0.1 |
| | | | Statistical significance of F1 | | | | |
| MOSI | 84.7±0.2 | 82.9±0.4 | 81.0±0.3 | 80.0±1.0 | 76.4±0.4 | 74.4±0.6 | 71.7±0.5 |
| MOSEI | 83.8±0.3 | 81.9±0.3 | 79.6±0.6 | 76.4±0.1 | 74.7±0.3 | 72.5±0.6 | 69.2±0.4 |
| IEMOCAP | 80.3±0.2 | 80.0±0.3 | 77.8±0.4 | 77.3±0.1 | 77.1±0.3 | 76.1±0.4 | 76.0±0.3 |

To better evaluate HME in real-world scenarios, we conduct two additional experiments. (i) Since real-world modality missing varies, we test HME using 10 different random seeds to simulate diverse missing modality conditions. (ii) Recognizing that the number of available samples in practical applications can vary, which affects the generation of hyper-modality representations, we test HME with 10 different batch sizes. The results of these experiments on the MOSI dataset, presented in Table 9, show that HME maintains strong stability and consistently delivers robust performance.

### A.3.4 Error Analysis

There may be failure cases where samples within a batch are semantically similar but express different sentiments. When these samples are used to enhance each other, they can introduce conflicting signals, a problem we call negative enhancement. To better understand the effect of negative enhancement, we conducted experiments on the MOSI dataset, focusing on cases where the cosine similarity between selected representations was greater than 0.9, to examine whether these representations aligned more closely with the ground-truth labels or with the model's predictions, and to identify the factors contributing to mis-classification.

Table 9: Testing stability under random missing protocol on MOSI (95% confidence intervals).

| | 0.1 | 0.2 | 0.3 | 0.4 | 0.5 | 0.6 | 0.7 |
|---|---|---|---|---|---|---|---|
| | Statistical significance with 10 different random seeds | | | | | | |
| ACC | 83.9±0.4 | 82.3±0.4 | 80.9±0.4 | 78.8±0.8 | 76.4±0.5 | 73.6±1.2 | 72.8±1.2 |
| F1 | 83.6±0.4 | 82.2±0.4 | 80.7±0.4 | 78.9±0.8 | 76.1±0.6 | 73.1±1.1 | 71.5±0.4 |
| | Statistical significance with 10 different batch sizes | | | | | | |
| ACC | 84.3±0.4 | 82.2±0.4 | 81.0±0.4 | 78.6±0.7 | 76.4±0.3 | 74.0±0.9 | 73.1±0.8 |
| F1 | 84.0±0.4 | 82.2±0.4 | 80.7±0.5 | 78.7±0.7 | 76.1±0.4 | 73.6±1.4 | 71.7±0.4 |

Table 10: Alignment analysis of high-similarity samples on MOSI.

| | Align with Ground Truth | Otherwise |
|---|---|---|
| Correct Predictions | 87.08% (5647/6485) | 12.92% (838/6485) |
| Wrong Predictions | 26.26% (255/971) | 73.74% (716/971) |

The results, summarized in Table 10, reveal two key findings. (i) The selected representations align more with the model's predictions than with the ground truth. Specifically, there are 6,363 cases where they match the predictions, compared to 5,902 cases where they match the ground truth. This suggests that semantically similar tones may bias the model toward incorrect sentiment predictions. (ii) In misclassified instances, the selected representations tend to reinforce the model's incorrect predictions rather than the true labels (716 vs. 255 cases). This highlights the influence of negative enhancement in amplifying model bias during classification.

To address this, HME applies two strategies. (i) During sample selection, a similarity threshold $t_s$ is introduced: only samples with feature similarity above this threshold are considered valid neighbors. If the similarity falls below the threshold, the model either uses the average batch representation or discards the neighbor entirely. This reduces the risk of injecting misleading information. (ii) The VIB module helps suppress residual noise. By enforcing compressed and task-relevant representations, VIB filters out inconsistencies and ensures that the fused features remain robust. Together, these strategies lead to consistent improvements across datasets and missing-modality scenarios.

### A.3.5 Generalization and Plug-and-Play Applicability Analysis

To evaluate the plug-and-play applicability of the proposed HME framework, we integrated the Hyper-Modality Representation Generation (HMG) module into the MPLMM architecture and tested it under random missing-modality conditions on the MOSI dataset. As shown in Table 11, incorporating HMG consistently improved performance, yielding average gains of +1.2 in accuracy and +1.8 in F1 score across different missing rates. These improvements highlight the plug-and-play compatibility of HME with existing models.

We further explored the generalization capability of HME from two perspectives: hyper-parameter sensitivity and cross-dataset transferability. First, we assessed the generalization performance by directly applying the hyper-parameters trained and optimized on MOSI dataset, to two external datasets, UR-FUNNY [45] and MUStARD [46]. As reported in Table 12, HME again outperformed the MPLMM baseline across multiple missing-modality configurations. This consistent improvement suggests that the hyper-parameters optimized on MOSI transfer well to other datasets, confirming the robustness and scalability of the HME.

Next, to test cross-dataset generalization ability, we trained and validated HME on one dataset and evaluated it on another using identical configurations. The results, summarized in Table 13, show a noticeable performance drop when transferring from MUStARD to UR-FUNNY, with average accuracy decreasing from 68.7 to 51.3 and F1 score from 69.9 to 55.5. In contrast, the reverse transfer showed a smaller degradation, with accuracy decreasing from 64.6 to 61.3 and F1 from 65.8 to 64.0. This can be attributed to inherent dataset differences and size disparities. MUStARD and UR-FUNNY differ substantially in modality composition and data distribution. For example, accuracy performance with all modalities ('L,A,V') present drops notably between the two datasets (75.0 vs. 49.7 and 72.6 v.s. 63.2). The presence of missing modalities further amplifies this domain gap. Moreover, the smaller size of MUStARD (539/68/68 for train/validation/test) provides limited variability compared to the much larger UR-FUNNY dataset (7614/980/994), making transfer from

Table 11: Ablation experiments of MPLMM on MOSI with random missing protocol.

| Methods | 0.1 | 0.2 | 0.3 | 0.4 | 0.5 | 0.6 | 0.7 | Avg. |
|---|---|---|---|---|---|---|---|---|
| Accuracy Performance | | | | | | | | |
| MPLMM | 81.4 | 78.7 | 74.7 | 70.3 | 67.1 | 62.5 | 61.0 | 70.8 |
| + HMG | 81.9 | 78.5 | 75.0 | 72.1 | 69.2 | 65.2 | 61.9 | 72.0 |
| Δ | +0.5 | -0.2 | +0.3 | +1.8 | +2.1 | +2.7 | +0.9 | +1.2 |
| F1 Performance | | | | | | | | |
| MPLMM | 81.4 | 78.7 | 74.8 | 69.9 | 66.3 | 60.5 | 58.3 | 70.0 |
| + HMG | 81.9 | 78.6 | 75.1 | 72.3 | 69.3 | 64.5 | 61.1 | 71.8 |
| Δ | +0.5 | -0.1 | +0.3 | +2.4 | +3.0 | +4.0 | +2.8 | +1.8 |

Table 12: Performance on UR-FUNNY and MUStARD datasets with fixed missing protocol.

| Methods | L | A | V | L,A | L,V | A,V | L,A,V | Avg. |
|---|---|---|---|---|---|---|---|---|
| Accuracy performance on UR-FUNNY | | | | | | | | |
| MPLMM | 67.6 | 61.8 | 61.8 | 67.6 | 67.6 | 61.8 | 67.6 | 65.1 |
| HME | 70.6 | 63.2 | 61.7 | 75.0 | 72.0 | 63.2 | 75.0 | 68.7 |
| F1 performance on UR-FUNNY | | | | | | | | |
| MPLMM | 68.1 | 61.8 | 61.8 | 68.1 | 68.1 | 61.8 | 68.1 | 65.4 |
| HME | 70.6 | 63.9 | 67.2 | 75.0 | 73.1 | 64.6 | 75.0 | 69.9 |
| Accuracy performance on MUStARD | | | | | | | | |
| MPLMM | 71.7 | 52.3 | 53.0 | 72.4 | 71.7 | 54.8 | 72.4 | 64.0 |
| HME | 72.9 | 53.3 | 52.5 | 72.8 | 72.9 | 55.5 | 72.6 | 64.6 |
| F1 performance on MUStARD | | | | | | | | |
| MPLMM | 71.8 | 54.3 | 54.9 | 72.5 | 71.8 | 55.2 | 72.5 | 64.7 |
| HME | 73.0 | 56.6 | 56.8 | 73.0 | 73.0 | 55.8 | 72.7 | 65.8 |

MUStARD to UR-FUNNY more challenging. Conversely, training on UR-FUNNY benefits from greater data diversity, leading to a smaller performance decline when transferring to MUStARD.

### A.3.6 VIB Components Analysis

To mitigate misleading semantic signals that may arise during cross-sample enhancement, HME includes a VIB module. As shown in Table 10, the retrieved similar samples occasionally contain sentiment-related cues that contradict the sentiment label of the target instance, thereby introducing noise and degrading model performance. The VIB module addresses this issue by compressing latent representations and retaining only task-relevant information. The ablation results presented in Tables

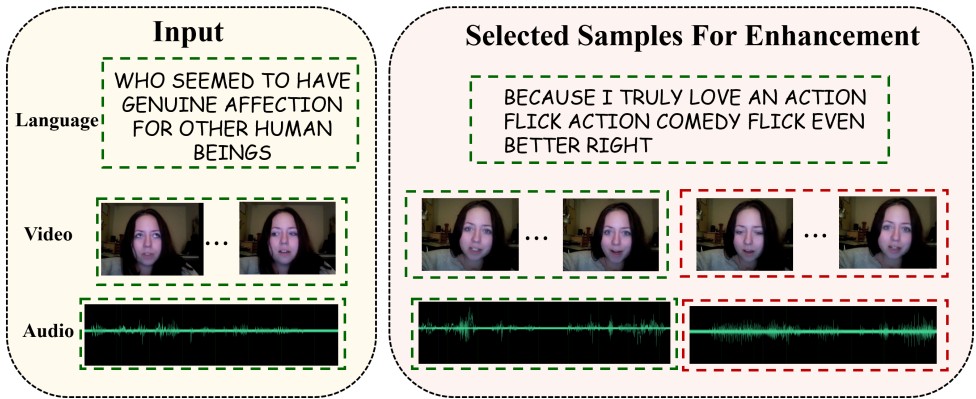

Figure 6: Case visualization of selected enhancement samples for each modality on the MOSI dataset. The similarity threshold is set to 0.9 and the batch size to 8. Green boxes indicate representations with the same label, while red boxes denote samples with different labels.

Table 13: Generalization performance between MUStARD and UR-FUNNY.

| Methods | L | A | V | L,A | L,V | A,V | L,A,V | Avg. |
|---|---|---|---|---|---|---|---|---|
| Accuracy Performance from MUStARD to UR-FUNNY | | | | | | | | |
| HME | 50.8 | 50.3 | 50.9 | 54.4 | 51.5 | 51.6 | 49.7 | 51.3 |
| F1 Performance from MUStARD to UR-FUNNY | | | | | | | | |
| HME | 52.5 | 52.9 | 52.5 | 56.2 | 55.5 | 53.6 | 65.4 | 55.5 |
| Accuracy Performance from UR-FUNNY to MUStARD | | | | | | | | |
| HME | 63.2 | 60.2 | 57.4 | 63.2 | 61.8 | 60.3 | 63.2 | 61.3 |
| F1 Performance from UR-FUNNY to MUStARD | | | | | | | | |
| HME | 66.3 | 62.9 | 60.2 | 66.3 | 64.7 | 60.4 | 67.1 | 64.0 |

Table 14: Performance comparison under different weights of VIB components.

| $I(Z,X)$ | 0.1 | 0.2 | 0.5 | 1.0 | 2.0 | 5.0 |
|---|---|---|---|---|---|---|
| HME | 78.0/75.9 | 77.4/75.3 | 76.8/74.9 | 76.5/73.7 | 76.0/72.8 | 75.5/71.3 |
| $I(Z,Y)$ | 0.1 | 0.2 | 0.5 | 1.0 | 2.0 | 5.0 |
| HME | 77.3/74.9 | 77.5/75.4 | 77.6/76.0 | 77.5/75.1 | 77.3/74.9 | 77.1/74.5 |
| VIB | 0.1 | 0.2 | 0.5 | 1.0 | 2.0 | 5.0 |
| HME | 77.7/75.1 | 77.8/75.5 | 77.6/76.0 | 76.9/75.0 | 76.5/74.4 | 75.7/73.8 |

6 and 7 indicate that removing the VIB module consistently results in performance degradation across all datasets, underscoring its effectiveness in suppressing misleading or noisy cues.

The VIB objective comprises two components, $I(Z,X)$ and $I(Z,Y)$, which quantify the amount of information retained from the input and the relevance of the latent representation to the target label, respectively. To assess the contribution of each component, we perform a detailed analysis by varying their relative weights both independently and jointly. As reported in Table 14, increasing the weight of $I(Z,X)$, which enforces stronger compression, tends to discard useful sentiment information and consequently reduces performance. In contrast, increasing the weight of $I(Z,Y)$ initially improves accuracy by enhancing label alignment, but excessive emphasis on this term ultimately leads to performance deterioration, likely due to overfitting or the under-utilization of complementary multimodal cues.

### A.3.7 Enhanced Representations Analysis

To examine the relationship between enhanced, raw, and selected-sample representations, we compute the mean squared error (MSE) between representations at different enhancement stages: modality enhancement ($E_m$), hyper-modality generation ($R_m$), noise-reduced compression ($F_m$), and modality fusion ($S_m$). Each is compared against the corresponding representation ($H_m$) and the average of selected samples ($H'_m$). Specifically, experiments are conducted on samples containing all three modalities ('L,A,V') with a similarity threshold of 0.9 on the MOSI training set. Table 15 reports the mean MSE across all modalities, and Table 16 focuses on the language modality.

The results show that enhanced representations ($E_m$, $R_m$, $F_m$) are consistently closer to $H'_m$ than to $H_m$, with MSE differences within 0.2. This indicates that enhancement effectively incorporates information from related samples while preserving original modality representation. After fusion, $S_m$ exhibits comparable MSE distances to both $H_m$ and $H'_m$, slightly favoring $H_m$. This suggests that the uncertainty-aware fusion mechanism prioritizes original representations while adaptively weighting enhanced features based on their variance.

Table 15: Mean MSE distance between different representations of three modalities on MOSI dataset.

| | $E_m$ | $R_m$ | $F_m$ | $S_m$ |
|---|---|---|---|---|
| $H_m$ | 3.61 | 3.76 | 2.93 | 0.76 |
| $H'_m$ | 3.47 | 3.63 | 2.79 | 0.77 |

Table 16: Mean MSE distance between language representations on MOSI dataset.

| | $E_L$ | $R_L$ | $F_L$ | $S_L$ |
|---|---|---|---|---|
| $H_L$ | 2.71 | 2.70 | 2.19 | 0.51 |
| $H'_L$ | 2.53 | 2.52 | 1.99 | 0.53 |

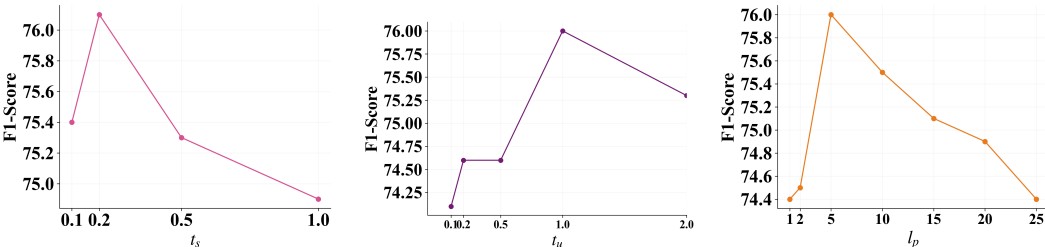

Figure 7: Performance of different $t_s$, $t_u$ and $l_p$ in IEMOCAP dataset with $MR = 0.7$.

Since the enhanced representations at each stage are obtained from the mean representation of the selected samples ($H'_m \in \mathbb{R}^d$), the attention weights of the learnable prompts cannot distinguish which selected sample contributes more. Therefore, we omit the visualization and analysis of this part and leave it for future work.

### A.3.8 Hyper-parameter Analysis

In this section, we present the analysis of hyper-parameters, including the similarity threshold ($t_s$), upper bound of uncertainty weights ($t_u$), and the length of the prompts ($l_p$). Figure 7 shows the variation in F1 score for HME under the random missing modality protocol ($MR = 0.7$) on the IEMOCAP dataset, with different values of $t_s$, $t_u$ and $l_p$.

**Effects of different $t_s$.** As $t_s$ increases, the F1 score initially improves, then starts to decline. The worst performance occurs when $t_s = 1.0$, where HME relies solely on the average representation. At $t_s = 0.1$, the performance is even better than at $t_s = 0.5$. This suggests that when fewer modality representations are available within the batch, lower values of $t_s$ allow for more diverse representations, even though they may have some error. The VIB module helps to extract useful information from these representations. As $t_s$ increases, the representations that meet the high similarity threshold become less frequent, reducing the amount of useful information, which leads to performance degradation.

**Effects of different $t_u$.** As $t_u$ increases, the F1 score shows a similar trend of initially increasing, then decreasing. The highest performance is observed at $t_u = 1.0$. This improvement can be attributed to the fact that, at MR = 0.7 (where only one modality is present per sample), many of the original representations in the fusion process are likely to be missing and replaced with zero vectors. A $t_u$ value of 1.0 helps to compensate for these missing values, providing sufficient information.

**Effects of different $l_p$.** As $l_p$ increases, the F1 score again shows an initial improvement followed by a decline. When $l_p$ is small, the model can only carry limited information, which restricts its performance. At $l_p = 5$, the model contains the most complete set of modality information, yielding the best performance. However, as $l_p$ increases further, the prompt may start to include unnecessary or irrelevant information, which can negatively affect the model's performance.

### A.3.9 Case Visualization and Representation Visualization

Here we visualize selected sample pairs from the MOSI dataset under different batch sizes. Specifically, we examine an extra configuration with a MR of 0.7, a similarity threshold of 0.9 and a batch size of 8 (Figure 6). From Figures 4 and 6, we observe that larger batch sizes increase the likelihood that the selected samples share the same sentiment label as the current input. When the batch size is large, HME benefits from a broader selection pool and tends to choose samples with labels consistent with the current input. In contrast, when the batch size is small and some modalities are missing, HME relies on the available representations for enhancement.

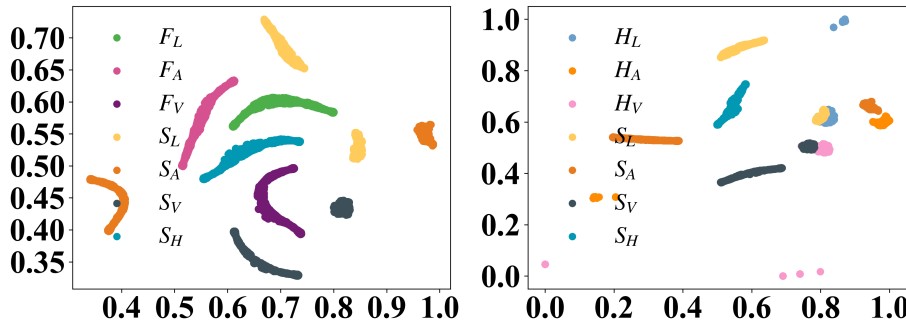

Figure 8: Visualization of learned representations on the MOSI dataset with $MR = 0.7$.

Table 17: Comparison of parameter count and running time for different models on the MOSI dataset.

| Models | # Parameters | running time ($\downarrow$) |
|---|---|---|
| MPLMM | 128,335,507 | 3,433s |
| DiCMoR | 113,007,057 | 12,792s |
| IMDer | 122,484,049 | 175,138s |
| HME | 118,542,893 | 684s |

Besides, to further understand the differences between the enhanced hyper-modality representations and the original representations, we visualize the additional representations using t-SNE: the original modality representation $H_m$, and the hyper-modality representation $F_m$, and the fused representations $S_m$ and $S_H$. Figure 8 shows these visualizations on the MOSI test set with an MR of 0.7. Our observations are as follows: (i) After uncertainty-aware attention fusion, the information from $F_m$ is well integrated into the original representation. This results in non-overlapping clusters between $S_m$, $S_H$ and $F_m$. (ii) The hyper-modality representation $F_m$ integrates effectively with the original modality representation $H_m$, as evidenced by overlapping information between the fused representations.

### A.3.10 Computation Overhead and Trade-off Analysis

**Computation overhead.** The computational efficiency of HME with three state-of-the-art baselines is compared on the MOSI dataset. To ensure fairness, all models were trained for 100 epochs under identical settings on an NVIDIA GTX 3090 GPU. As shown in Table 17, HME completed training in just 684 seconds, which is only 20% of the runtime of MPLMM, 5% of DiCMoR, and less than 1% of IMDer. In terms of model size, HME has approximately 118M parameters, slightly more than DiCMoR but fewer than most other baselines. These results highlight the notable efficiency of HME in both training speed and parameter count.

**Trade-off between performance and computational cost.** Since HME's enhanced representations are highly sensitive to batch size, larger batches may improve performance but require greater computational resources. To examine this trade-off and explore the performance upper bound, we conducted additional experiments on MOSI under the random missing protocol. Specifically, the average accuracy and F1 score with varying the batch size across seven missing rates is evaluated. As summarized in Table 18, performance improves steadily as batch size increases, peaking at 256.

Table 18: Computational overhead of HME with different batch sizes on the MOSI dataset.

| Batch Size | ACC/F1 | Training Time | Peak GPU Memory |
|---|---|---|---|
| 64 | 76.6/76.3 | 429s | 5.6GB (5,635MB) |
| 128 | 77.3/77.0 | 315s | 8.3GB (8,279MB) |
| 256 | 79.0/78.7 | 290s | 13.2GB (13,169MB) |
| 288 | 77.8/77.4 | 280s | 14.6GB (14,621MB) |
| 320 | 78.1/77.6 | 280s | 15.8GB (15,785MB) |
| 352 | 77.9/77.2 | 273s | 17.0GB (17,023MB) |

Beyond this point, both accuracy and F1 score show a slight decline. This suggests that while larger batches generally benefit learning, there is a saturation point beyond which gains diminish and minor degradation may occur.

To provide a more practical perspective, we also measured training time (over 50 epochs) and peak GPU memory usage under different batch sizes in Table 18. The results reveal three trends: (i) training time decreases as batch size grows, which may due to fewer updates per epoch, (ii) GPU memory usage increases, and (iii) performance improves up to batch size 256, then gradually declines. Overall, a batch size of 256 offers the best balance—delivering peak performance with acceptable training time and manageable memory consumption.

