# OpenReview forum: "Hyper-Modality Enhancement for Multimodal Sentiment Analysis with Missing Modalities"
_NeurIPS.cc/2025/Conference — NeurIPS 2025 poster_

### Official Review · Reviewer_bLRy · 2025-06-30

**Clarity:** 3
**Significance:** 3
**Originality:** 3
**Rating:** 5
**Confidence:** 3

**Summary:**

This paper proposes a novel framework, Hyper-Modality Enhancement (HME), for Multimodal Sentiment Analysis (MSA) in the presence of missing modalities. The method begins by extracting modality-specific representations, then enhances them using batch-level cross-sample information via a modality enhancement module. These enhanced features are integrated through a hyper-modality generation module that fuses cues across all modalities. To address redundancy and noise, the framework incorporates a Variational Information Bottleneck (VIB). Experimental results on multiple benchmark datasets demonstrate that HME outperforms prior methods, especially under high missing rates, showing strong robustness and generalization.

**Questions:**

1. Experimental setting clarification: It is unclear whether the missing modality setting is consistent across training and testing. For example, when evaluating under “L missing”, is the language modality also missing during training, or only during inference? This clarification is important for interpreting the robustness of the model.

5. Clarification of Figure 3b (Section 4.4): The authors state that the enhanced hyper-modality representations capture more sentiment-relevant information, forming distinct clusters in the t-SNE plot. Could you elaborate more on this point?

**Ethical Concerns:**

["NO or VERY MINOR ethics concerns only"]

**Final Justification:**

The rebuttal addressed all my questions and the experiments are well presented. I have raised my rating.

**Quality:**

3

**Strengths And Weaknesses:**

Strengths:

1. The paper tackles the underexplored challenge of missing modalities in MSA using a cross-sample enhancement mechanism, which introduces complementary information from semantically similar samples in the same batch—without relying on full modality reconstruction.

2. The uncertainty-aware fusion strategy is thoughtfully designed, using modality-wise variance to weight the contribution of each representation. The accompanying qualitative analysis of uncertainty weights provides insightful understanding of how different modalities contribute under varying conditions.

Weakness:

1. Batch size sensitivity: Since enhanced features are derived from samples within the same batch, it would be valuable to analyze how performance varies with batch size. This could help understand the dependency of HME on batch-level diversity. Qualitative examples would be helpful to understand why certain samples were chosen for enrichment.

2. Interpretation of prompts: The paper could benefit from further analysis of the learnable prompts used in the Perceiver and hyper-modality interaction modules. What information is being captured or attended to by these prompts?

3. More detailed ablation on VIB: While VIB is shown to reduce redundancy, a more granular study of the influence of the VIB loss weight $\alpha$ would be useful. Additionally, a comparison of performance with and without VIB (especially under different MR values) would clarify its specific contribution.

---

> ### Author Rebuttal · Authors · 2025-07-30
>
> We sincerely thank Reviewer bLRy for your time and insightful feedback. We are especially grateful for your recognition of the thoughtful design of our proposed framework. Your comments have significantly helped us deepen our analysis and improve the clarity of our paper. Below, we address each of your concerns in detail.
> ### **Weaknesses & Questions:**
>
> **W1: Batch Size Sensitivity & Qualitative Examples of Enhancement**
>
> (1) Thank you for highlighting the potential impact of batch size on our HME. We agree that this is a crucial aspect that can offer valuable insights into HME’s robustness and practical deployment. To explore this, we conducted experiments on the MOSI dataset under the random missing protocol, evaluating batch sizes of 64, 128, and 256. The results are summarized in Tables 1 and 2 below (to be included in the appendix). We observed that larger batch sizes consistently lead to better performance. This is likely because a larger batch offers more diverse and semantically similar samples, allowing the enhancement module to extract richer information. These findings will be included in the revised appendix.
>
> Table 1: Accuracy performance on MOSI with random missing protocol
> | batch size | 0.1  | 0.2  | 0.3  | 0.4  | 0.5  | 0.6  | 0.7  | avg  |
> | ---------- | ---- | ---- | ---- | ---- | ---- | ---- | ---- | ---- |
> | 64         | 83.8 | 81.1 | 79.1 | 76.1 | 74.2 | 72.3 | 69.7 | 76.6 |
> | 128        | 83.8 | 81.4 | 79.9 | 76.7 | 76.5 | 72.4 | 70.3 | 77.3 |
> | 256        | 84.9 | 82.9 | 81.1 | 79.9 | 76.4 | 74.5 | 73.5 | 79.0 |
>
> Table 2: F1 performance on MOSI with random missing protocol
> | batch size | 0.1  | 0.2  | 0.3  | 0.4  | 0.5  | 0.6  | 0.7  | avg  |
> | ---------- | ---- | ---- | ---- | ---- | ---- | ---- | ---- | ---- |
> | 64         | 83.8 | 81.2 | 78.6 | 76.1 | 73.9 | 71.2 | 69.2 | 76.3 |
> | 128        | 83.8 | 81.4 | 79.6 | 76.3 | 75.9 | 72.1 | 69.7 | 77.0 |
> | 256        | 84.7 | 82.9 | 81.0 | 80.0 | 76.4 | 74.4 | 71.7 | 78.7 |
>
> (2) As for qualitative examples of enhancement, We appreciate your suggestion to include more qualitative examples. While Figure 6 in appendix currently shows some case visualizations from the MOSI dataset, we agree that it would be beneficial to expand this analysis. In the revised version, we will include visualizations under varying batch sizes and similarity thresholds, highlighting which samples are selected for enhancement and why. This will provide readers with a more concrete grasp of how and why enhancement contributes to performance.
>
> **W2: Interpretation of Learnable Prompts**
>
> Thank you for your insightful comment regarding the prompts used in the Perceiver and hyper-modality interaction modules. Inspired by your suggestion, we realized that simple visualization (as shown in Figures 3b and 5) is insufficient to fully convey their function. We will enhance our qualitative analysis by including:
> - Visualization of how changing the similarity threshold affects prompt behavior and sample selection,
> - Analysis of attention weights assigned to each prompt during enhancement,
> - Measurements of distances between original, enhanced, and hyper-modality embeddings.
>
> These analyses will clarify the function and utility of prompts in attending to sentiment-relevant information and will help distinguish the role of hyper-modality from raw or unimodal representations.
>
> **W3: Detailed Ablation on the Variational Information Bottleneck (VIB)**
>
> We thank you for encouraging a deeper investigation into the role of the VIB module. Following your suggestion, we conducted a more granular analysis of  $I(Z,X)$ and $I(Z,Y)$. Specifically, we varied their weights individually and jointly. As shown in Table 3 below, increasing the weight of $I(Z,X)$ compresses representations too aggressively, degrading performance by removing sentiment-relevant features. Conversely, increasing $I(Z,Y)$ improves performance up to a point, after which it declines—likely due to overfitting or underemphasizing alignment with input features. Jointly increasing both follows a similar trend.
>
> Table 3: Performance comparison under different weights of VIB components
> |   I(Z,X)   |    0.1    |    0.2    |    0.5    |     1     |     2     |     5     |
> | :--------: | :-------: | :-------: | :-------: | :-------: | :-------: | :-------: |
> |    HME     | 78.0/75.9 | 77.4/75.3 | 76.8/74.9 | 76.5/73.7 | 76.0/72.8 | 75.5/71.3 |
> | **I(Z,Y)** |  **0.1**  |  **0.2**  |  **0.5**  |   **1**   |   **2**   |   **5**   |
> |    HME     | 77.3/74.9 | 77.5/75.4 | 77.6/76.0 | 77.5/75.1 | 77.3/74.9 | 77.1/74.5 |
> |  **VIB**   |  **0.1**  |  **0.2**  |  **0.5**  |   **1**   |   **2**   |   **5**   |
> |    HME     | 77.7/75.1 | 77.8/75.5 | 77.6/76.0 | 76.9/75.0 | 76.5/74.4 | 75.7/73.8 |
>
> Additionally, to further demonstrate the effectiveness of VIB, we provide an ablation study comparing HME with and without the VIB component across various missing rates and datasets, as shown in Tables 6 and 7 in the appendix. These results consistently show performance drops when VIB is removed, underscoring its contribution to robustness and representation quality.
>
> **Q1: Experimental Setting Clarification**
>
> Thank you for pointing this out. We clarify that in the _fixed missing modality_ setting (e.g., when “L” is missing), the modality is only removed during validation and inference—not during training. This setup aligns with prior works (GCNet (TPAMI 2023), MPLMM (ACL 2024), DiCMoR (ICCV 2023), IMDer (NeurIPS 2023), and LNLN (NeurIPS 2024)) and is intended to simulate real-world scenarios where certain modalities may be missing only at test time. In contrast, under the _random missing protocol_, the same missing rate is applied uniformly during training, validation, and testing. We will revise the paper to state this more explicitly.
>
> **Q2: Clarification of Figure 3b (Section 4.4)**
>
> We apologize for the lack of clarity in our explanation. Figure 3b, when considered alongside Figure 5 (in the appendix), provides a more complete picture. In Figure 3b, the t-SNE plot shows that the hyper-modality representations ($F_m$) form more distinct sentiment-aware clusters than the original unimodal representations ($H_m$), indicating that $F_m$ captures more discriminative information. Figure 5 further demonstrates that after applying uncertainty-aware fusion, the $F_m$ representation is effectively integrated with the original $H_m$, resulting in more structured and separable clusters. Together, these visualizations support our claim that the hyper-modality enhances sentiment-relevant information in a meaningful way.
>
> Once again, we sincerely thank the reviewer for your detailed and thoughtful feedback, which has helped us improve both the depth and clarity of our paper. We will incorporate these enhancements into our revision.

---

> > ### Author Response · Authors · 2025-08-07
> > **Response to Reviewer bLRy**
> >
> > We sincerely apologize for any inconvenience caused by this follow-up. As the rebuttal deadline approaches, we would like to kindly check whether our earlier response has addressed your concerns. If there are still any aspects that remain unclear or could benefit from further clarification, we would be more than happy to provide additional explanations or make further improvements while time allows. Thank you again for your time, thoughtful comments, and consideration.

---

### Official Review · Reviewer_py7h · 2025-07-01

**Clarity:** 3
**Significance:** 4
**Originality:** 3
**Rating:** 4
**Confidence:** 3

**Summary:**

In this paper, authors introduce Hyper-Modality Enhancement (HME), a novel framework for robust multimodal sentiment analysis under missing modality conditions. By leveraging cross-sample semantic cues to enrich observed modalities without explicit reconstruction, HME addresses limitations of prior methods that rely on pseudo-missing simulations or teacher-student frameworks. The approach comprises three key components: (1) a hyper-modality generation module that distills sentiment-relevant information from other samples via learnable prompts and variational information bottleneck (VIB) for noise filtering, (2) an uncertainty-aware fusion mechanism that adaptively balances original and enhanced representations based on representation reliability, and (3) extensive experiments demonstrating consistent performance gains over SOTA methods on CMU-MOSI, CMU-MOSEI, and IEMOCAP datasets under various missing modalities.

**Questions:**

1. Authors claim VIB reduces noise and redundancy, but it’s unclear how VIB interacts with the uncertainty-aware fusion mechanism. For example:
(1)Does VIB’s compression degrade important sentiment signals?
(2)How does the balance between compression (I(Z,X)) and informativeness (I(Z,Y)) affect performance?

2. Experiments focus on in-dataset missingness (fixed/random masks), but real-world missingness often arises from domain shifts (e.g., sensor failures in deployed systems). How would HME perform on datasets with:
(1)Natural missingness (e.g., Twitter posts with missing images)?
(2)Cross-dataset evaluation (e.g., training on CMU-MOSI and testing on IEMOCAP)?

3. Why use cosine similarity for cross-sample selection? Are there alternatives (e.g., contrastive learning)?

4. How does the uncertainty weighting (Eq. 5) relate to uncertainty quantification in Bayesian deep learning?

5.What are the ethical risks of inferring emotions from incomplete data (e.g., misrepresentation of marginalized groups)?

**Ethical Concerns:**

["NO or VERY MINOR ethics concerns only"]

**Limitations:**

Yes

**Quality:**

3

**Strengths And Weaknesses:**

Strengths :
The content of this paper presents a technically sound and well-motivated framework for robust multimodal sentiment analysis under missing modalities:
(1) HME addresses key limitations of prior work by avoiding explicit modality reconstruction and instead leveraging cross-sample semantic cues through learnable prompts and a variational information bottleneck (VIB).
(2) Three-module design, modality enhancement, hyper-modality generation, and uncertainty-aware fusion, is conceptually clear and addresses both noise reduction and representation enrichment.
(3) The experimental evaluation is thorough, demonstrating consistent improvements over strong baselines (e.g., IMDer, LNLN) across three datasets (CMU-MOSI, CMU-MOSEI, IEMOCAP) and varying missing-modality scenarios.

Weakness:
Technical contributions are solid, but the paper’s evaluation has limitations that temper its impact.
(1) Experiments focus primarily on within-dataset generalization under controlled missingness patterns (fixed/random missing protocols), but do not explore domain shifts or out-of-distribution robustness.
(2) Theoretical underpinnings of certain design choices, such as the similarity threshold (tₛ) for cross-sample selection and the interplay between VIB compression and uncertainty weighting, are not fully justified.
(3) Finally, while the visualizations (e.g., t-SNE plots) aid interpretability, the paper could better articulate how hyper-modality representations qualitatively differ from raw features and why this distinction matters for sentiment prediction.

---

> ### Author Rebuttal · Authors · 2025-07-30
>
> We sincerely thank Reviewer py7h for the time and effort dedicated to reviewing our paper. We deeply appreciate your recognition of our work as "a technically sound and well-motivated framework", and we value the insightful comments and constructive suggestions you provided. Below, we respond to each of your concerns in detail.
> ### **Weaknesses & Questions:**
>
> **W2&Q1: Interaction Between VIB and Uncertainty-Aware Fusion**
>
> Thank you for highlighting this important aspect. Your comments helped us further explore the interplay between the variational information bottleneck (VIB) and the uncertainty-aware fusion.
>
> (1) The VIB module is specifically introduced to filter out misleading semantic signals that may appear during cross-sample enhancement. For example, as shown in **Table 10 (Appendix)**, the selected similar samples may contain sentiment-relevant cues that contradict the target sample’s sentiment label, leading to performance degradation. Rather than aggregating all available signals, VIB helps retain only reliable features. The ablation results (**Tables 6 and 7 in Appendix**) show that removing VIB results in a consistent performance drop across all datasets, indicating its effectiveness in suppressing noise and misleading cues.
>
> (2) Motivated by your question on the trade-off between $I(Z,X)$ and $I(Z,Y)$, we conducted a detailed study by varying their weights individually and jointly. Results (see Table 1 below) show that increasing the $I(Z,X)$ weight (i.e., stronger compression) tends to suppress useful sentiment information, degrading performance. Conversely, while increasing $I(Z,Y)$ initially improves results, excessive emphasis eventually hurts performance—likely due to under-utilization of complementary features or overfitting.
>
> Table 1: Performance comparison under different weights of VIB components
> |   I(Z,X)   |    0.1    |    0.2    |    0.5    |     1     |     2     |     5     |
> | :--------: | :-------: | :-------: | :-------: | :-------: | :-------: | :-------: |
> |    HME     | 78.0/75.9 | 77.4/75.3 | 76.8/74.9 | 76.5/73.7 | 76.0/72.8 | 75.5/71.3 |
> | **I(Z,Y)** |  **0.1**  |  **0.2**  |  **0.5**  |   **1**   |   **2**   |   **5**   |
> |    HME     | 77.3/74.9 | 77.5/75.4 | 77.6/76.0 | 77.5/75.1 | 77.3/74.9 | 77.1/74.5 |
> |  **VIB**   |  **0.1**  |  **0.2**  |  **0.5**  |   **1**   |   **2**   |   **5**   |
> |    HME     | 77.7/75.1 | 77.8/75.5 | 77.6/76.0 | 76.9/75.0 | 76.5/74.4 | 75.7/73.8 |
>
> **W1&Q2: Evaluation Under Real-World Missingness and Cross-Domain Settings**
>
> We agree that assessing performance under natural missingness and domain shifts is crucial for real-world applicability. Due to the lack of standard datasets with naturally missing modalities (e.g., social media posts with missing images or audio), we followed existing work (IMDer (NeurIPS 2023) and LNLN (NeurIPS 2024)) in simulating missingness. However, HME differs fundamentally by not relying on ground-truth representations of the missing modalities, aiming for more realistic simulation.
>
> For domain-shifts scenarios, we have extended our experiments to scenarios using the MUStARD and UR-FUNNY sarcasm detection datasets. Results (Tables 2–5 below) show that HME consistently outperforms baseline across various missing-modality conditions, demonstrating improved robustness in domain-shift settings.
>
> As for cross-dataset generalization, we faced challenges due to label inconsistency (e.g., binary vs. four-class sentiment in MOSI/MOSEI vs. IEMOCAP) and differing input dimensions (e.g., (5/20 vs. 74/35) of audio/video features in MOSI vs. MOSEI). Nonetheless, we acknowledge its importance and will extend our experiments using the MUStARD and UR-FUNNY sarcasm detection datasets. We will include these findings and additional experiments in our revised manuscript.
>
> Table 2: Accuracy performance on MUStARD with fixed missing protocol
> | methods | L    | A    | V    | LA   | LV   | AV   | LAV  | avg  |
> | ------- | ---- | ---- | ---- | ---- | ---- | ---- | ---- | ---- |
> | MPLMM   | 67.6 | 61.8 | 61.8 | 67.6 | 67.6 | 61.8 | 67.6 | 65.1 |
> | HME     | 70.6 | 63.2 | 61.7 | 75.0 | 72.0 | 63.2 | 75.0 | 68.7 |
>
> Table 3: F1 performance on MUStARD with fixed missing protocol
> | methods | L    | A    | V    | LA   | LV   | AV   | LAV  | avg  |
> | ------- | ---- | ---- | ---- | ---- | ---- | ---- | ---- | ---- |
> | MPLMM   | 68.1 | 61.8 | 61.8 | 68.1 | 68.1 | 61.8 | 68.1 | 65.4 |
> | HME     | 70.6 | 63.9 | 67.2 | 75.0 | 73.1 | 64.6 | 75.0 | 69.9 |
>
> Table 4: Accuracy performance on UR-FUNNY with fixed missing protocol
>
> | methods | L    | A    | V    | LA   | LV   | AV   | LAV  | avg  |
> | ------- | ---- | ---- | ---- | ---- | ---- | ---- | ---- | ---- |
> | MPLMM   | 71.7 | 52.3 | 53.0 | 72.4 | 71.7 | 54.8 | 72.4 | 64.0 |
> | HME     | 72.9 | 53.3 | 52.5 | 72.8 | 72.9 | 55.5 | 72.6 | 64.6 |
>
> Table 5: F1 performance on UR-FUNNY with fixed missing protocol
>
> | methods | L    | A    | V    | LA   | LV   | AV   | LAV  | avg  |
> | ------- | ---- | ---- | ---- | ---- | ---- | ---- | ---- | ---- |
> | MPLMM   | 71.8 | 54.3 | 54.9 | 72.5 | 71.8 | 55.2 | 72.5 | 64.7 |
> | HME     | 73.0 | 56.6 | 56.8 | 73.0 | 73.0 | 55.8 | 72.7 | 65.8 |
>
> **Q3: Choice of cosine similarity for cross-sample selection**
>
> Thank you for this valuable suggestion. We chose cosine similarity due to its robustness to curse of dimensionality and its widespread use in related work. Moreover, many contrastive learning methods—including CLIP—also rely on cosine similarity as a similarity metric.
>
> In our case, cross-sample selection must be unsupervised, since real-world missingness often occurs in settings without label access. Unlike contrastive learning approaches that require carefully curated positive/negative pairs or rely on known modality alignment (as in CLIP), HME is designed to flexibly select multiple relevant samples without supervision, improving adaptability across diverse applications.
>
> **Q4: Relationship between Eq. 5 and Bayesian Deep Learning (BDL) uncertainty**
>
> Equation 5 draws inspiration from uncertainty quantification principles in BDL, particularly in its use of variance to estimate uncertainty. However, while BDL typically estimates model uncertainty via techniques like Monte Carlo dropout, our method estimates uncertainty at the representation level—specifically, the robustness of VIB-compressed features. Thus, Eq. 5 shares a conceptual link with BDL, but it serves a distinct purpose in modulating the fusion of original and enhanced modalities based on feature reliability.
>
> **Q5: Ethical risks of emotion inference from incomplete data**
>
> We fully acknowledge the ethical implications of inferring emotions from incomplete inputs. Inaccurate emotion recognition can lead to misrepresentation, particularly of marginalized groups, or raise concerns about profiling and privacy. While our method aims to reduce such risks by improving robustness to missing modalities, we recognize the importance of transparent and responsible deployment. We will explicitly discuss these concerns and potential safeguards in our revised manuscript.
>
> **W3: Visualizing hyper-modality representations**
>
> Thank you for pointing out the need for clearer qualitative explanations of our hyper-modality representations. While we include relevant visualizations (e.g., **Figure 3b**, **Figure 5**, and **Figure 6** in the appendix), we agree that these could be more illustrative.
>
> To address this, we plan to extend our qualitative analysis in the revision, including:
> - Visualization of how varying the similarity threshold affects selected samples,
> - Analysis of attention weights during enhancement,
> - Measurement of distance between original, enhanced, and hyper-modality embeddings.
> This will help demonstrate how hyper-modality differs meaningfully from raw or enhanced representations and why that distinction is important for sentiment understanding.
>
> We deeply appreciate your thoughtful feedback, which has significantly improved the clarity and depth of our work. We are enthusiastic about further enhancing our manuscript based on your suggestions and thank you again for your constructive review.

---

> > ### Comment · Reviewer_py7h · 2025-08-06
> >
> > Thank you for sharing the detailed rebuttal draft. I went through it carefully and I think the responses are clear, well-structured, and persuasive.
> >
> > For W2 & Q1, I find the explanation of the role of VIB convincing. The ablation evidence and the additional analysis of weight trade-offs make the argument solid.
> >
> > For W1 & Q2, the extension to MUStARD and UR-FUNNY is a strong addition, and your discussion of the challenges in cross-dataset settings is reasonable and transparent.
> >
> > For Q3–Q4, the rationale for cosine similarity and the clarification of the link (and distinction) between Eq. 5 and BDL are logical and well put.
> >
> > For Q5 & W3, I appreciate that you openly acknowledge ethical considerations and propose further visualization/qualitative analysis — this strengthens the work’s clarity and responsibility.
> >
> > Lastly, I feel the rebuttal is well-prepared, addresses the reviewer’s comments thoroughly, and is ready to go. I’m okay with this version.

---

> > > ### Author Response · Authors · 2025-08-07
> > > **Response to Reviewer py7h**
> > >
> > > Thank you sincerely for your thoughtful and constructive feedback. We are encouraged to know that our clarifications and additional analyses helped address your concerns and contributed to a clearer understanding of our work. Your recognition of the revisions is both reassuring and motivating for us. If the clarifications above fully address your concerns, we would sincerely appreciate it if you could reflect this in your rating. Thank you again for your engagement and for helping us enhance the contribution of our research.

---

### Official Review · Reviewer_QBVh · 2025-07-02

**Clarity:** 3
**Significance:** 3
**Originality:** 3
**Rating:** 5
**Confidence:** 4

**Summary:**

The paper is focused on the task of multimodal sentiment analysis with missing modalities. To address this task, the authors propose an approach, called Hyper-Modality Enhancement, which leverages cross-sample information to enrich each observed modality with semantic information from other samples and thus prevent performance degradation when some data modalities are missing. Furthermore, the approach uses an uncertainty-aware fusion mechanism that balances the original and enriched representations to improve model robustness. Experimental results on several benchmark datasets show that the proposed approach performs better than the baselines considered under several missing modality settings.

**Questions:**

Are there any existing multimodal datasets that have missing modalities already, so that one does not have to simulate that using the fixed missing and random missing protocols?

Is the fixed missing protocol used here for the first time? What is the motivation for it?

**Ethical Concerns:**

["NO or VERY MINOR ethics concerns only"]

**Final Justification:**

I had several questions for the authors. They satisfactorily answered my questions. The inclusion of those clarifications in the final version of the paper would be useful.

**Limitations:**

The authors discuss several limitations.

**Paper Formatting Concerns:**

No concerns

**Quality:**

3

**Strengths And Weaknesses:**

The approach proposed is intuitive and at the same time original. The two modules are well motivated and the authors do a good job at both providing the big picture and explaining the details.

The authors compare the performance of the proposed approach with multiple existing baselines on multiple datasets. Experimental results show that the proposed approach performs better than the baselines under various missing modality settings.

Ablation studies show that all components of the proposed approach contribute to the performance of the model.

There are a few components of the model that rely on prior works and approaches. The authors could provide some additional details about those approaches in the Appendix, if not in the main manuscript, to make it clearer what is novel to their approach. Otherwise, the proposed approach may be seen as rather incremental.

The authors evaluate the baseline models by comparison with the proposed approach under two missing modality protocols (fixed missing and random missing). At least one of the protocols (random missing) has been used in prior works. The authors used existing implementations to get results for the baselines. It is not clear if the exact data used in prior works (with missing modalities) was not available or why the authors were not able to compare directly with results from prior works. It may be that the results of the baselines are not fully optimized, although the authors followed the specific guidelines from prior works. Also, the authors did not justify well the fixed missing protocol, which may not have been used in prior works.

---

> ### Author Rebuttal · Authors · 2025-07-30
>
> We sincerely thank Reviewer QBVh for the time and effort invested in reviewing our paper, as well as for the constructive feedback provided. We are particularly encouraged that you found our approach both intuitive and original. Below, we address each of your comments in detail.
> ### **Weaknesses & Questions:**
>
> **W1: Clarification of Components Based on Prior Work**
>
> Thank you for pointing out the need to clarify which components of our HME build on prior work. We agree that distinguishing the novel aspects of our approach is important for transparency. In response, we will add more details to the Appendix—such as a clearer description of frameworks like Perceiver—to better contextualize our contributions. This will help readers more easily identify what is new in our HME.
>
> **Q1: On Existing Datasets with Naturally Missing Modalities**
>
> We appreciate your insightful question regarding datasets that contain naturally missing modalities. To the best of our knowledge, such multimodal sentiment datasets are currently unavailable. As a result, the standard practice across the literature—including the baseline methods we compare against—is to simulate missing modalities by selectively dropping them from fully observed datasets. One of the key distinctions of our HME, is that it does not rely on the ground-truth representations of missing modalities during simulation. This design choice aims to more closely reflect real-world conditions and enhance the practical relevance of our method.
>
> **Q2: Motivation for the Fixed Missing Protocol**
>
> Thank you for raising the question regarding the fixed missing protocol. This is not the first time such a protocol has been used. Notable examples include baseline models like DiCMoR (ICCV 2023), IMDer (NeurIPS 2023), and LNLN (NeurIPS 2024). The motivation behind the fixed missing setting is to simulate realistic scenarios where specific modalities may be consistently unavailable—for example, when an audio sensor fails and only textual and visual information is recorded. Evaluating under this protocol helps better understand how models perform under modality-specific failures that could occur in deployment environments.
>
> **W2: Clarification on Baseline Results**
>
> We also appreciate your concern regarding the reproducibility and optimization of baseline results. In some cases, we were unable to directly compare with results reported in prior work under identical settings—either because the specific missing modality protocol was not used, or the results for certain datasets (such as IEMOCAP) were not reported for those conditions. In such cases, we carefully followed the official implementations and experimental guidelines provided by the original authors to ensure fair and meaningful comparisons. The implementation details are shown in Lines 459-464 in appendix: "To identify the hyper-parameter of the re-implemented baselines, we adhered to the hyper-parameter guidelines provided in the official implementations of these models. For models like IMDer and DiCMoR, which included pre-trained weights, we used the publicly available checkpoints and configurations. For baselines without available hyper-parameters for specific datasets, we aligned their hyper-parameters with HME’s settings where possible. When unique parameters were required, we referred to their recommended settings from other publicly available implementations."
>
> Once again, we are grateful for your thoughtful feedback, which has helped us further improve the clarity and rigor of our work.

---

> > ### Comment · Reviewer_QBVh · 2025-08-05
> >
> > Thank you for answering my questions. Including some of these clarifications in the paper would be useful.

---

> > > ### Author Response · Authors · 2025-08-07
> > > **Response to Reviewer QBVh**
> > >
> > > We sincerely thank the reviewer QBVh for the valuable feedback. In the revised version, we will incorporate the relevant clarifications into the main text and appendix to improve the paper’s clarity.

---

### Official Review · Reviewer_5Y4d · 2025-07-03

**Clarity:** 3
**Significance:** 3
**Originality:** 2
**Rating:** 4
**Confidence:** 4

**Summary:**

The authors propose Hyper-Modality Enhancement (HME)  that addresses the challenge of missing modalities in real-world scenarios. Instead of reconstructing missing modalities for MM sentiment analysis. The main ideas is that HME enriches missing modalities with semantically relevant information from other samples in the dataset. The main components are a Hyper-Modality Representation Generation Module that uses cross-sample enhancement and bottleneck for noise reduction, and an uncertainty-based fusion module that weights modality contribution based on variance estimates.

**Questions:**

1. How do you compare with SOTA for your baseline system?
2. Is your method applicable to other / any MSA architecture?
3. Variance based fusion is very flaky. Can you quantify your claims that for missing modalities your variance estimates are more robust?
4. It would be interesting to note how your hyper-parameters transfer to a new database?
5. How much data to you really need to identify relevant samples and to tune your hyperparameters?
6. How would your approach work with more tightly integrated MM MSA frameworks, e.g., https://arxiv.org/pdf/2504.11082 or https://lab-msp.com/MSP-Podcast_Competition/IS2025/

[I will revisit my review after some of these questions and issues are clarified by the authors]

**Ethical Concerns:**

["NO or VERY MINOR ethics concerns only"]

**Final Justification:**

The extensive additional experiments and clarifications significantly strengthen the paper's contributions and practical applicability.

**Limitations:**

yes

**Quality:**

3

**Strengths And Weaknesses:**

Strengths

1.  Addresses an interesting and practical problem in a novel way: the cross-sample enhancement strategy is innovative leveraging information from semantically similar samples across the dataset.
2. Good evaluation section: Extensive experiments on three benchmark datasets (CMU-MOSI, CMU-MOSEI, IEMOCAP) with both fixed and random missing protocols. Thorough analysis.
3. Good result: HME achieves notable performance gains  - albeit there is quite a bit of tuning going on
4. Uncertainty-aware fusion (while not very novel) seem to fit well in this framework
5. The details in the appendix are much appreciated.


Weaknesses

1. Unclear to me how the baseline system compares to SOTA.
2. Not clear how the proposed method could be plug n play to any MSA architecture.
3. Scalability/performance concerns to a new dataset with very limited data and unknown distribution
4. Hyperparameter Sensitivity: The method introduces several hyperparameters (similarity threshold, uncertainty bounds etc.) that that appear to require careful tuning for different datasets (see appendix they seem very spiky)

---

> ### Author Rebuttal · Authors · 2025-07-30
>
> We sincerely thank Reviewer 5Y4d for the time and thoughtful feedback provided. We are encouraged by your recognition that our work addresses a practical and important problem with a novel cross-sample enhancement strategy. Below, we respond point-by-point to your concerns and suggestions.
> ### **Weaknesses & Questions:**
>
> **W1&Q1: Comparison with State-of-the-Art (SOTA) Methods**
>
> Thank you for highlighting the need for clarification. Our baseline systems were selected from the latest state-of-the-art methods specifically designed for multimodal sentiment analysis under missing modality scenarios. These include GCNet (TPAMI 2023), MPLMM (ACL 2024), DiCMoR (ICCV 2023), IMDer (NeurIPS 2023), and LNLN (NeurIPS 2024). We will clearly state this selection criterion in the revised manuscript to avoid confusion and better contextualize our comparisons.
>
> **W2&Q2: Generalizability and Plug-and-Play Applicability of HME**
>
> We appreciate your suggestion regarding the plug-and-play applicability of HME. This inspired us to further evaluate the modularity of our HME framework. To demonstrate its generality, we integrated our Hyper-Modality Representation Generation (HMG) module into the MPLMM architecture and evaluated it under random missing protocols on the MOSI dataset. As shown in Tables 1 and 2 below, incorporating HMG improved performance in most situations, with average accuracy and F1 gains of 1.2 and 1.8 across various missing rates. These results demonstrate the plug-and-play compatibility of HME with existing architectures and confirm its general effectiveness. We will include these findings in our revised paper.
>
> Table 1: Accuracy performance on MOSI with random missing protocol
> | methods  | 0.1  | 0.2  | 0.3  | 0.4  | 0.5  | 0.6  | 0.7  | avg  |
> | -------- | ---- | ---- | ---- | ---- | ---- | ---- | ---- | ---- |
> | MPLMM    | 81.4 | 78.7 | 74.7 | 70.3 | 67.1 | 62.5 | 61.0 | 70.8 |
> | +HMG     | 81.9 | 78.5 | 75.0 | 72.1 | 69.2 | 65.2 | 61.9 | 72.0 |
> | $\Delta$ | +0.5 | -0.2 | +0.3 | +1.8 | +2.1 | +2.7 | +0.9 | +1.2 |
>
> Table 2: F1 performance on MOSI with random missing protocol
> | methods  | 0.1  | 0.2  | 0.3  | 0.4  | 0.5  | 0.6  | 0.7  | avg  |
> | -------- | ---- | ---- | ---- | ---- | ---- | ---- | ---- | ---- |
> | MPLMM    | 81.4 | 78.7 | 74.8 | 69.9 | 66.3 | 60.5 | 58.3 | 70.0 |
> | +HMG     | 81.9 | 78.6 | 75.1 | 72.3 | 69.3 | 64.5 | 61.1 | 71.8 |
> | $\Delta$ | +0.5 | -0.1 | +0.3 | +2.4 | +3.0 | +4.0 | +2.8 | +1.8 |
>
> **Q3: Robustness of Variance-Based Fusion**
>
> We conducted ablation studies to validate the contribution of the variance-based uncertainty fusion module. Specifically, we removed the uncertainty weighting (denoted as w/o $U_m$​) and observed performance drops across all datasets and settings, which is detailed in Tables 6 and 7 in appendix, confirming that our uncertainty estimates improve the model’s robustness in handling missing modalities.
>
> **W3&W4&Q4: Hyperparameter Sensitivity and Generalization to New Datasets**
>
> Thank you for pointing out the concern regarding hyperparameter sensitivity and potential challenges in generalizing to new datasets. Inspired by your suggestion, we conducted additional experiments on two external datasets—**UR-FUNNY** and **MUStARD**—using the **same** hyperparameters originally tuned on MOSI. As reported in Tables 3 through 6 (to be included in the revised manuscript), **HME consistently outperformed the MPLMM baseline across various missing modality settings**, even without re-tuning. These results suggest that the proposed hyperparameters generalize reasonably well across datasets, demonstrating both the **scalability** and **robustness** of our method.
>
> Table 3: Accuracy performance on MUStARD with fixed missing protocol
> | methods | L    | A    | V    | LA   | LV   | AV   | LAV  | avg  |
> | ------- | ---- | ---- | ---- | ---- | ---- | ---- | ---- | ---- |
> | MPLMM   | 67.6 | 61.8 | 61.8 | 67.6 | 67.6 | 61.8 | 67.6 | 65.1 |
> | HME     | 70.6 | 63.2 | 61.7 | 75.0 | 72.0 | 63.2 | 75.0 | 68.7 |
>
> Table 4: F1 performance on MUStARD with fixed missing protocol
> | methods | L    | A    | V    | LA   | LV   | AV   | LAV  | avg  |
> | ------- | ---- | ---- | ---- | ---- | ---- | ---- | ---- | ---- |
> | MPLMM   | 68.1 | 61.8 | 61.8 | 68.1 | 68.1 | 61.8 | 68.1 | 65.4 |
> | HME     | 70.6 | 63.9 | 67.2 | 75.0 | 73.1 | 64.6 | 75.0 | 69.9 |
>
> Table 5: Accuracy performance on UR-FUNNY with fixed missing protocol
> | methods | L    | A    | V    | LA   | LV   | AV   | LAV  | avg  |
> | ------- | ---- | ---- | ---- | ---- | ---- | ---- | ---- | ---- |
> | MPLMM   | 71.7 | 52.3 | 53.0 | 72.4 | 71.7 | 54.8 | 72.4 | 64.0 |
> | HME     | 72.9 | 53.3 | 52.5 | 72.8 | 72.9 | 55.5 | 72.6 | 64.6 |
>
> Table 6: F1 performance on UR-FUNNY with fixed missing protocol
> | methods | L    | A    | V    | LA   | LV   | AV   | LAV  | avg  |
> | ------- | ---- | ---- | ---- | ---- | ---- | ---- | ---- | ---- |
> | MPLMM   | 71.8 | 54.3 | 54.9 | 72.5 | 71.8 | 55.2 | 72.5 | 64.7 |
> | HME     | 73.0 | 56.6 | 56.8 | 73.0 | 73.0 | 55.8 | 72.7 | 65.8 |
>
> **Q5: Data Requirements for Sample Retrieval and Hyperparameter Tuning**
>
> Thank you for raising this important question. We address it from two perspectives:
> - **Relevant Sample Selection**: The number of semantically similar samples selected by the HME is closely tied to the batch size used during training. To provide a concrete analysis, we examined this on the MOSI dataset under a missing rate (MR) of 0.3. The Table 7 below summarizes the average number of selected relevant samples (with cosine similarity > 0.9) and the corresponding model performance. Even with a batch size of 64, HME is able to **retrieve nearly 8 highly similar samples** on average, which **already results in competitive performance**. At a batch size of 128, HME outperforms all baselines, and with 256, further gains are observed. These results demonstrate that HME does not require a large batch size to find meaningful enhancements—moderate-sized batches suffice.
> - **Hyperparameter Tuning**: Hyperparameter tuning is performed using the validation set. Taking MOSI as an example, the validation set contains 229 samples. With a batch size of 128 during validation, the effective tuning is conducted on batches of 128. We applied grid search for key hyperparameters using this relatively small validation set. Despite the limited size, **the resulting configuration generalized well across both validation and test scenarios**, as shown in our main experiments.
> In conclusion, HME is able to identify relevant samples and perform effective tuning without requiring large-scale data. This aligns well with real-world constraints and supports the practicality of our HME.
>
> Table 7: Performance and average number of selected samples of HME in MOSI with MR=0.3
> | batchsize | Avg samples | Acc/F1    |
> | --------- | ----------- | --------- |
> | 64        | 7.84        | 79.1/78.6 |
> | 128       | 13.38       | 79.9/79.6 |
> | 256       | 25.15       | 81.1/81.0 |
>
> **Q6: Integration with More Tightly Coupled MSA Frameworks**
>
> We appreciate your references to more tightly integrated multimodal frameworks (e.g., DeepMLF, MSP-Podcast Challenge). HME can be flexibly incorporated into these frameworks in three ways:
> 1. **Representation-Level Integration**: HME can be applied after unimodal encoders to enhance feature robustness before fusion. Our integration with MPLMM already validates this design.
> 2. **Fusion-Level Integration**: Many tightly coupled models still treat all modalities equally. Our uncertainty-aware fusion can dynamically adjust weights based on modality reliability, which is especially useful given the noisier nature of audio and video compared to text.
> 3. **Broader Applicability to Missing Modalities**: These frameworks often assume all modalities are available. HME enables them to operate in **real-world missing-modality scenarios**, extending their practical utility.
> We will elaborate on these integration strategies and provide implementation guidance in the updated manuscript.
>
> Thank you once again for your constructive feedback. Your comments have helped us strengthen both the clarity and generalizability of our work. We hope our responses address your concerns and demonstrate the potential of HME to serve as a robust and flexible solution in real-world scenarios.

---

> > ### Comment · Reviewer_5Y4d · 2025-08-06
> > **Reply to Authors**
> >
> > Thank you for the comprehensive responses and additional experiments. I particularly appreciate the plug-and-play demonstration with MPLMM and the cross-dataset validation using the same hyperparameters from MOSI, addressing my concerns about generalizability. The additional experiments and clarifications strengthen the paper's contributions. I will raise my score by 1.

---

> > > ### Author Response · Authors · 2025-08-07
> > > **Response to Reviewer 5Y4d**
> > >
> > > We are delighted that the additional experiments addressing HME's generalizability were helpful and met your expectations. Thank you once again for your thoughtful suggestions, which led us to explore deeper aspects of HME’s robustness and broaden the scope of our evaluation.

---

### Official Review · Reviewer_Robh · 2025-07-14

**Clarity:** 3
**Significance:** 2
**Originality:** 2
**Rating:** 4
**Confidence:** 4

**Summary:**

This paper introduces Hyper-Modality Enhancement (HME) framework for Multimodal Sentiment Analysis which is robust against missing modalities. The main contribution is a two-stage pipeline. First, the "Hyper-Modality Representation Generation" module enriches the representation of each available modality by borrowing semantically relevant information from other samples within the same training batch. This cross-sample enhancement reduces the model's reliance on having complete data for every sample. Second, an "Uncertainty-Aware Fusion" mechanism adaptively combines the original modality representations with these newly enhanced ones. It uses Variational Information Bottleneck (VIB), allowing the model to dynamically trust the enhanced features more or less depending on their estimated reliability.

**Questions:**

- The cross-sample enhancement is a good idea but is dependent on the quality of neighbors in the batch. Have you considered strategies to mitigate the risk of "negative enhancement" from misleadingly similar samples?
- How does the selection of batch size impact the performance?
- As mentioned in line 135-137, "If no representation satisfies the threshold $t_s$, or if the modality is missing, we instead average all available representations of the current modality within the batch to capture the modality-specific information" - What is the reason behind this assumption? If I understood it correctly, the samples are not very similar and thus they can be from different classes. How does averaging them provide an "Enhanced" representation?
- How does the total number of parameters and number of learnable parameters compare with other methods?

**Ethical Concerns:**

["NO or VERY MINOR ethics concerns only"]

**Final Justification:**

I appreciate the authors’ detailed response and the clarifications provided, as well as the additional experimental data supporting their claims. However, I find that certain limitations and trade-offs between computational cost and performance remain and warrant more thorough analysis. After reviewing the explanation and conducting my own assessment, I have decided to maintain my current rating.

**Limitations:**

Yes

**Quality:**

3

**Strengths And Weaknesses:**

**Strengths:**
- The proposed method looks technically sound and well-motivated. The architecture thoughtfully combines several concepts to enhance robustness to missing modalities.
- The empirical evaluation is thorough and convincing.
- The paper is well-written, structured, and easy to follow.

**Weaknesses:**
- The core enhancement mechanism relies on finding similar samples within a mini-batch. This makes the model's performance potentially sensitive to batch composition and size.
- The pairwise similarity calculation within a batch has a computational complexity of $O(B^2)$, where $B$ is the batch size. While this is acceptable for the batch sizes used in the paper, it could become a bottleneck for training with very large batches, limiting scalability.
- The methodology section is presented logically but seemed a little bit overloaded with symbols and equations inside paragraphs. Careful rewriting will enhance readability.

---

> ### Author Rebuttal · Authors · 2025-07-30
>
> We sincerely thank Reviewer Robh for the time and effort dedicated to reviewing our paper and for the insightful comments and constructive suggestions. We are encouraged by the positive feedback, especially the recognition that our method is "technically sound and well-motivated." Below, we address the concerns and questions raised in detail.
>
> ### **Weaknesses & Questions:**
>
> **W1&Q2: Sensitivity to Batch Composition and Size**
>
> We appreciate your observation regarding the model’s sensitivity to batch composition and size. To analyze this, we evaluated the performance of HME under different batch sizes ({64, 128, 256}) on the MOSI dataset using the random missing modality protocol. The results in Table 1 and Table 2 below show that although performance slightly varies with batch size, HME consistently outperforms baseline methods. Specifically, as batch size increases, both accuracy and F1-score improve, suggesting that more diverse batches help identify better semantic neighbors.
>
> Table 1: Accuracy performance on MOSI with random missing protocol
> | batch size | 0.1  | 0.2  | 0.3  | 0.4  | 0.5  | 0.6  | 0.7  | avg  |
> | ---------- | ---- | ---- | ---- | ---- | ---- | ---- | ---- | ---- |
> | 64         | 83.8 | 81.1 | 79.1 | 76.1 | 74.2 | 72.3 | 69.7 | 76.6 |
> | 128        | 83.8 | 81.4 | 79.9 | 76.7 | 76.5 | 72.4 | 70.3 | 77.3 |
> | 256        | 84.9 | 82.9 | 81.1 | 79.9 | 76.4 | 74.5 | 73.5 | 79.0 |
>
> Table 2: F1 performance on MOSI with random missing protocol
> | batch size | 0.1  | 0.2  | 0.3  | 0.4  | 0.5  | 0.6  | 0.7  | avg  |
> | ---------- | ---- | ---- | ---- | ---- | ---- | ---- | ---- | ---- |
> | 64         | 83.8 | 81.2 | 78.6 | 76.1 | 73.9 | 71.2 | 69.2 | 76.3 |
> | 128        | 83.8 | 81.4 | 79.6 | 76.3 | 75.9 | 72.1 | 69.7 | 77.0 |
> | 256        | 84.7 | 82.9 | 81.0 | 80.0 | 76.4 | 74.4 | 71.7 | 78.7 |
>
> These experiments will be added to the appendix to clarify this sensitivity.
>
> **W2: Scalability and Pairwise Similarity Overhead**
>
> You're absolutely right that computing pairwise similarity within the batch introduces a computational overhead, especially as batch size grows. While larger batches improve the chance of finding semantically similar representations, they also increase computation. We acknowledge this trade-off and plan to explore more scalable solutions in future work, such as approximate neighbor search or memory-efficient batch construction strategies.
>
> **W3: Overloaded Mathematical Presentation**
>
> Thank you for pointing out the readability issue in Sections 3.3 and 3.4. We will revise the presentation by simplifying the notation and separating key equations from the main text for better clarity. For instance, auxiliary variables such as $\sigma + \epsilon$ will be redefined or renamed to reduce reading load for readers.
>
> **Q1: Risk of Negative Enhancement**
>
> We greatly appreciate this insightful observation. Indeed, as you pointed out, semantically similar but sentimentally dissimilar samples within a batch can introduce conflicting signals—what is referred to as “negative enhancement.” In Section A.2.4 and Table 10 of the appendix, we conduct a detailed analysis of this issue. Empirically, we observed that around 12.9% of selected neighbors in a batch could have sentiment labels opposite to the target sample, which may lead to unintended distortion of the enhanced representation.
>
> To mitigate this, (1) we implement **a similarity threshold** during the neighbor selection phase. Specifically, only samples whose feature similarity exceeds a threshold are considered valid neighbors. This helps **reduce the likelihood of injecting misleading information**. When the similarity falls below this threshold, the model either defaults to using the average batch representation or excludes the neighbor entirely. In addition, (2) the VIB module plays a crucial role in further **suppressing noise introduced by unreliable enhancements**. By promoting compressed and task-relevant representations, the VIB mechanism helps **filter out semantic inconsistencies** and ensures that the fused features remain robust. We have observed that incorporating this strategy, especially with the VIB component, **improves performance across datasets and missing-modality scenarios** (as shown in Tables 6 and 7 of the appendix). We will revise the main manuscript to explicitly describe this mechanism and include additional analysis to highlight its effectiveness.
>
> Furthermore, inspired by  frameworks such as MoCo, we are exploring the use of a memory bank to retrieve reliable neighbors from a broader pool of samples beyond a single batch. This approach could further reduce the risk of negative enhancement and is a promising direction for future work.
>
> **Q3: Averaging Modality Representations When No Good Matches Exist**
>
> Thank you for this excellent question. In scenarios where no high-similarity representations are found or when a modality is entirely missing, we fall back on averaging all available representations of that modality within the batch. While these samples may come from different classes, this fallback provides a modality-level common representation that helps stabilize performance and avoid drastic degradation. Furthermore, VIB plays a crucial role in reducing irrelevant or noisy information during fusion, helping minimize the negative impact of such averaging.
>
> **Q4: Parameter Count Comparison**
>
> We appreciate the suggestion to report model complexity. As shown in Table 3 below, we report the parameter count and runtime comparison on the MOSI dataset. HME has approximately 118M parameters—slightly more than DiCMoR (113M), but fewer than most other baselines. Importantly, HME is also efficient in runtime. HME requires only 684 seconds to train, significantly less than DiCMoR’s 12,792 seconds. This reflects the practical efficiency of our approach despite the enhanced representational capacity.
>
> Table 3:Parameter count comparison on MOSI dataset
> | Model   | Parameters      | Runtime (s) |
> | ------- | --------------- | ----------- |
> | MPLMM   | 128,335,507     | 3,433s      |
> | DiCMoR  | 113,007,057     | 12,792s     |
> | IMDER   | 122,484,049     | 175,138s    |
> | **HME** | **118,542,893** | **684s**    |
>
> Once again, we sincerely thank the reviewer for the thoughtful and constructive feedback. We are committed to improving the clarity, robustness, and scalability of HME, and we will incorporate all the suggested revisions in our updated manuscript and supplementary material.

---

> > ### Comment · Reviewer_Robh · 2025-08-05
> > **Follow-up questions**
> >
> > I thank the authors for their detailed and thoughtful response. I have a few follow-up questions and suggestions for clarification:
> > * As the experimental results indicate, the performance is sensitive to batch size and tends to improve as the batch size increases. Is there an upper bound to this improvement?
> > * The authors have acknowledged that increasing the batch size introduces additional computational complexity. A discussion on the trade-off between performance gains and computational cost would be valuable for readers.
> > * Regarding the statement: *“While these samples may come from different classes, this fallback provides a modality-level common representation that helps stabilize performance and avoid drastic degradation.”* Could the authors provide either theoretical justification or experimental evidence to support this claim?
> >
> > Addressing these points would further enhance the paper's contribution and accessibility to a broader audience.

---

> > > ### Author Response · Authors · 2025-08-06
> > > **Response to follow-up questions**
> > >
> > > We sincerely thank the reviewer Robh for the thoughtful follow-up questions and constructive suggestions. These questions have helped us further analyze the performance characteristics and limitations of our proposed HME model. Below, we provide detailed responses to each point.
> > >
> > > ### **Questions:**
> > >
> > > **Q1: Is there an upper bound to the performance gains from increasing batch size**
> > >
> > > Thank you for this insightful question. To explore the performance upper bound, we conducted additional experiments on the MOSI dataset under the random missing protocol. Specifically, we evaluated the average accuracy and F1 score across seven different MRs, using a range of batch sizes.
> > >
> > > As shown in **Table 1**, HME achieves the best performance when the batch size is set to **256**. Increasing the batch size beyond this point (e.g., to 288, 320, or 352) leads to a slight decrease in both accuracy and F1 score. This indicates that while performance generally improves with batch size, there is a saturation point beyond which further increases no longer yield benefits and may introduce minor degradation.
> > >
> > > **Table 1: Performance and Computational Overhead of HME on MOSI**
> > > | Batch Size |  Accuracy / F1  | Training Time (50 epochs) |  Peak GPU Memory   |
> > > | :--------: | :-------------: | :-----------------------: | :----------------: |
> > > |     64     |   76.6 / 76.3   |           429 s           |  5.6 GB (5635 MB)  |
> > > |    128     |   77.3 / 77.0   |           315 s           |  8.3 GB (8279 MB)  |
> > > |    256     | **79.0 / 78.7** |           290 s           | 13.2 GB (13169 MB) |
> > > |    288     |   77.8 / 77.4   |           280 s           | 14.6 GB (14621 MB) |
> > > |    320     |   78.1 / 77.6   |           280 s           | 15.8 GB (15785 MB) |
> > > |    352     |   77.9 / 77.2   |           273 s           | 17.0 GB (17023 MB) |
> > >
> > > **Q2: Trade-off between performance and computational cost**
> > >
> > > We appreciate your suggestion to analyze the trade-off between performance improvements and computational cost. We evaluated this trade-off by measuring model performance (accuracy and F1), training time (over 50 epochs), and peak GPU memory usage under different batch sizes. These results are also summarized in **Table 1**.
> > >
> > > We observe that as the batch size increases:
> > > - **Training time decreases** (fewer updates per epoch),
> > > - **GPU memory usage increases**, and
> > > - **Performance improves up to a point** (batch size = 256), then slightly declines.
> > > At a batch size of **256**, HME achieves the best trade-off—delivering peak performance with acceptable training time and manageable memory consumption. We will incorporate this analysis into the revised manuscript to provide readers with practical guidance regarding resource-performance trade-offs.
> > >
> > > **Q3: Justification for using representations from different classes to enhance missing modalities**
> > >
> > > We thank the reviewer for highlighting this important point, which significantly improves the interpretability of the HME. Our design assumes that representations from other samples, which are potentially from different classes, can provide useful information. To support this claim, we conducted ablation studies that remove modality-level enhancement for individual modalities. The results, presented in **Table 2 in main manuscript** and **Tables 6 and 7 in appendix**, show that removing any single enhanced representation consistently leads to performance drops. For instance, under the _random missing protocol_ with **MR = 0.7**, removing a single modality enhancement decreases accuracy from **73.5** to **≤70.7**, and F1 score from **71.7** to **≤70.0**. These results demonstrate that even when augmented from different-class samples, the additional information contributes to improved robustness and stabilization.
> > >
> > > Please let us know if additional clarification or further experimentation would be helpful. We deeply appreciate your continued engagement and valuable feedback.

---

> > > > ### Comment · Reviewer_Robh · 2025-08-06
> > > >
> > > > I thank the authors for the detailed response and clarification provided. After reviewing the explanation and conducting my own assessment, I have decided to maintain my current rating.

---

> > > > > ### Author Response · Authors · 2025-08-07
> > > > > **Response to Reviewer Robh**
> > > > >
> > > > > We sincerely thank the reviewer Robh once again for the thoughtful and constructive suggestions provided throughout the review process. Your feedback has been instrumental in deepening the technical quality of our work and refining its overall presentation. Although we regret that our clarifications did not lead to a change in your rating, we fully respect your decision and value your critical perspective. Incorporating your suggestions has significantly strengthened the clarity of our paper, making it more relevant to a broader audience. If the clarifications above fully address your concerns, we would sincerely appreciate it if you could reflect this in your rating. Thank you again for your engagement and for helping us enhance the contribution of our research.

---

### Note · Authors · 2025-08-11

We sincerely thank the Reviewers and the Area Chair for their time, effort, and thoughtful feedback on our work. We are encouraged by the positive remarks, including that the paper is “**technically sound and well-motivated**” (Reviewers Robh and py7h), “**addresses an interesting and practical problem in a novel way**” (Reviewer 5Y4d), “**intuitive and at the same time original**” (Reviewer QBVh), and “**thoughtfully designed**” (Reviewer bLRy).

Our work introduces **Hyper-Modality Enhancement (HME)**, a framework that enriches each observed modality with semantically relevant cues retrieved from other samples, rather than reconstructing missing modalities directly. This cross-sample enrichment reduces dependence on fully observed data during training, making it well suited to real-world scenarios with incomplete inputs. We also propose an uncertainty-aware fusion mechanism that adaptively balances original and enriched representations, improving robustness under different patterns of missingness.

In the rebuttal, we addressed the main concerns raised by the reviewers, including:
- The effect of batch size on performance and computation (Reviewers Robh, bLRy)
- Generalization to new datasets and integration with existing models (Reviewers 5Y4d, py7h)
- The impact of the VIB component’s loss weights on overall performance (Reviewers py7h, bLRy)

These were resolved through thorough analyses, additional experiments across batch sizes, datasets, and model architectures, as well as clearer explanations of key model components.

We are pleased that Reviewer Robh **maintained a positive score** after our clarifications, and that Reviewer 5Y4d **acknowledged their concerns on generalizability were resolved and expressed intent to raise their score**. Reviewer QBVh appreciated our clarifications and suggested incorporating some into the final paper. Reviewer py7h described our rebuttal as “**well-prepared, addresses the reviewer’s comments thoroughly, and is ready to go**”. For Reviewer bLRy, we believe we have fully addressed the issues raised, and we hope our additional explanations and results resolve any remaining concerns.

In summary, we are grateful for the constructive dialogue, which has strengthened our work and highlighted its novelty, soundness, and practical value. HME is applicable to existing models, demonstrates strong generalization across datasets, and delivers robust performance when inputs are incomplete.

---

### Decision · Program_Chairs · 2025-09-17

**Decision:**

Accept (poster)

**Comment:**

Summary. This paper introduces Hyper-Modality Enhancement (HME) framework for Multimodal Sentiment Analysis, which is robust against missing modalities. The main contribution is a two-stage pipeline: (1) a "Hyper-Modality Representation Generation" module enriches the representation of each available modality by borrowing semantically relevant information from other samples within the same training batch. (2) an "Uncertainty-Aware Fusion" mechanism adaptively combines the original modality representations with these newly enhanced ones.

Strengths. The proposed method looks technically sound and well-motivated. The empirical evaluation is thorough and convincing. The paper is well-written, structured, and easy to follow.

Weaknesses. The method seems sensitive to batch size and other parameters. Authors plan to clarify this in the final version.

The paper received 5,5,4,4,4 ratings. Authors submitted a rebuttal to reviewer comments and reviewers remained positive ratings.

Overall, a good paper with interesting results for robust multimodal sentiment analysis. We expect that authors will revise the paper according to reviewer comments and include all the promised changes in the final version.